# END-TO-END CONFORMAL PREDICTION FOR TRAJECTORY OPTIMIZATION

## ABSTRACT

Conformal Prediction (CP) is a powerful tool to construct uncertainty sets with coverage guarantees, which has fueled its extensive adoption in generating prediction regions for decision-making tasks, e.g., Trajectory Optimization (TO) in uncertain environments. However, existing methods predominantly employ a sequential scheme, where decisions rely unidirectionally on the prediction regions, and consequently the information from the decision-making end fails to be transmitted back to instruct the CP end. In this paper, we propose a novel End-to-End CP (E2E-CP) framework for shrinking-horizon TO with a joint risk constraint over the entire mission time. Specifically, a CP-based posterior risk calculation method is developed by fully leveraging the realized trajectories to adjust the posterior allowable risk, which is then allocated to future times to update prediction regions. In this way, the information in the realized trajectories is continuously fed back to the CP end, enabling attractive end-to-end adjustments of the prediction regions and a provable online improvement in trajectory performance. Furthermore, we theoretically prove that such end-to-end adjustments consistently maintain the coverage guarantees of the prediction regions, thereby ensuring provable safety. Additionally, we develop a decision-focused iterative risk allocation algorithm with theoretical convergence analysis for allocating the posterior allowable risk which closely aligns with E2E-CP. The effectiveness and superiority of the proposed method are demonstrated through benchmark experiments.

## 1 INTRODUCTION

In recent years, Trajectory Optimization (TO) has achieved significant success in fields such as autonomous driving Zhou et al. (2020), autonomous surface vessels Tsolakis et al. (2024), and coverage control Davis et al. (2016). However, collision-free TO in uncertain environments is a formidable challenge, because the intentions of obstacles are unknown. A crucial aspect of collision avoidance involves predicting obstacle trajectories. Existing trajectory prediction tools are unable to predict fully accurate trajectories. Therefore, a common approach is to generate the $(1-\alpha)$-coverage prediction regions of the obstacle trajectories. If these regions contain the true trajectories with a probability of at least $1 - \alpha$, they are considered *valid*. The key to probabilistic collision-free TO lies in adjusting the prediction regions while remaining valid to improve the trajectory performance.

Conformal Prediction (CP) is an attractive framework to produce prediction regions with finite-sample guarantees of validity Vovk et al. (2005); Shafer & Vovk (2008). Without imposing any assumptions about prediction algorithms and data distributions, CP utilizes a calibration dataset to obtain a valid prediction region for test data. Owing to its simplicity and versatility, CP and its variants have been widely applied in various safety-critical applications, such as probabilistic collision-free TO Lindemann et al. (2023); Sun et al. (2024), reliable estimation of graph neural networks H. Zargarbashi et al. (2023) and language modeling Quach et al. (2024).

However, there is a disconnect between existing research on CP theory and CP application for decision-making. On the side of CP theory, most existing work primarily focuses on upstream data, developing new CP algorithms to enhance prediction performance, such as addressing distributional shifts Gibbs & Candes (2021), performing multi-step time forecasting Sun & Yu (2023), and improving the efficiency of prediction regions Bai et al. (2022). There is a lack of CP algorithms focused on enhancing the performance of downstream decisions. On the side of CP application for

decision-making, most existing work embeds the CP into decision-making pipelines as a method for generating prediction regions, and employs a sequential approach, i.e. the prediction region is first computed using CP and then the decision depends unidirectionally on the prediction region without considering the favorable impact of the decision on the prediction region. However, this information channel blockage from the decision-making end to the CP end seriously prevents the CP from leveraging the information of past decisions to boost the performance of future decisions. Therefore, there is a pressing research need to develop an end-to-end framework that seamlessly integrates CP with decision-making, fully exploiting the information of past decisions to adjust prediction regions in an end-to-end fashion and thereby remarkably enhance the performance of future decisions.

To fill the aforementioned research gap, we propose an End-to-End CP (E2E-CP) framework for shrinking-horizon TO in uncertain environments and the collision risk over the total mission time is constrained at all times. The proposed framework leverages CP to construct the prediction regions of obstacle positions and adjusts these regions online in an end-to-end fashion while ensuring coverage guarantees, i.e. validity. In particular, we propose a novel posterior probability calculation method to obtain the posterior probability of collision conditional on realized trajectories. The posterior collision probability is then used to adjust the allowable collision risk, which is allocated to future times to yield prediction regions. In this manner, information from past trajectories is transmitted to the CP end through the posterior probability calculation, guiding the end-to-end adjustments of the prediction regions. Such adjustments in E2E-CP not only offer provable performance improvements but also consistently maintain the validity of the prediction regions. With the adjusted prediction regions, the trajectory is obtained by solving the resulting TO problem. Additionally, we further propose a decision-focused risk allocation method, i.e. Iterative Risk Allocation (IRA), which aims to optimize the trajectory performance by iteratively allocating the allowable risk to future times while enjoying the convergence guarantee. We highlight the main contributions of our work below.

- We propose, for the first time in the literature E2E-CP, a general uncertainty quantification framework closely associated with downstream decision-making which enables the adjustment of prediction regions using the feedback information embedded in decisions.

- We prove that 1) the end-to-end adjustments in E2E-CP do not compromise the coverage guarantees of prediction regions, and 2) E2E-CP offers guarantees for decision-making performance improvement. In other words, E2E-CP enjoys both validity and performance.

- We propose a decision-focused risk allocation algorithm with theoretical convergence analysis for E2E-CP, which optimizes the risk allocation to enhance decision-making performance.

## 2 RELATED WORK

**Conformal Prediction.** Conformal prediction originated in the early work Vovk et al. (1999; 2005); Shafer & Vovk (2008) to generate the prediction region. The salient advantage of CP lies in its ability to offer coverage guarantees regardless of prediction algorithms and data distributions. Most recently, various variants of CP have been developed to handle upstream data with different characteristics H. Zargarbashi et al. (2024); Liu et al. (2024) or to produce prediction regions in a wide array of forms Angelopoulos et al. (2024); Auer et al. (2023). In response to the distribution shift in the upstream data, ACI Gibbs & Candes (2021); Podkopaev et al. (2024); Zaffran et al. (2022) and EnbPI Xu & Xie (2021; 2023) developed CP through online learning and sliding window, respectively, and achieved asymptotic validity. In the context of multi-step time series forecasting, Sun & Yu (2023) combined CP with copula to propose the CopulaCPTS, while Cleaveland et al. (2024) employed an optimization-based method. Zhou et al. (2024b) presented a new conformal method for time series forecasting. In addition, numerous studies focused on improving the efficiency of the prediction region by changing the region shape Xu et al. (2024), minimizing the region length Kiyani et al. (2024), or directly optimizing the region construction function Bai et al. (2022). Note that the prediction regions are typically utilized by downstream tasks in a sequential manner. However, the aforementioned research work primarily aims to enhance the predictive performance of CP rather than directly improving the performance of downstream decision-making.

**TO in Uncertain Environments.** The probabilistic collision-free TO in uncertain environments relies on the accurate description of uncertainties. Robust optimization and chance-constrained optimization are typically employed in TO to mitigate collision risks Kuwata & How (2010); Petrović et al. (2022); Zhu & Alonso-Mora (2019). However, in these methods, the bound or distribution of

uncertainty is assumed to be perfectly available to construct the confidence set of uncertainty. Fortunately, the recent development of CP theory relaxes the above limitations and offers distribution-free methods for constructing the confidence set. Lindemann et al. (2023) and Strawn et al. (2023) applied CP to the safe planning for single-robot systems, while Muthali et al. (2023) and Kuipers et al. (2024) extended it to multi-robot systems. Additionally, Dixit et al. (2023) and Zhou et al. (2024a) employed the ACI to address the obstacle trajectory distribution shift. Stamouli et al. (2024) proposed a novel nonconformity score for shrinking-horizon TO. All the above methods directly employ CP in a sequential way to generate prediction regions. Nevertheless, the performance of realized trajectories has yet to be conveyed to the upstream CP end as feedback information to adjust the prediction regions, which has the great potential to further boost the performance of trajectory.

## 3 PROBLEM FORMULATION AND BACKGROUND

### 3.1 PROBLEM FORMULATION

Consider a discrete-time nonlinear dynamical system as follows.

$$x_{t+1} = f(x_t, u_t), \quad x_0 = x_{init} \tag{1}$$

where $x_t \in \mathcal{X} \subseteq \mathbb{R}^{n_x}$ and $u_t \in \mathcal{U} \subseteq \mathbb{R}^{n_u}$ are the state and control at time $t = 0, ..., T$, respectively, and $T \geq 1$ is the total mission time. The sets $\mathcal{U}$ and $\mathcal{X}$ represent the admissible sets of control inputs and system states, respectively. The function $f : \mathbb{R}^{n_x} \times \mathbb{R}^{n_u} \to \mathbb{R}^{n_x}$ represents the system dynamics and $x_{init}$ is the initial state of the system. For brevity, let $x_{t_1:t_2} := (x_{t_1}, ..., x_{t_2})$ and $u_{t_1:t_2} := (u_{t_1}, ..., u_{t_2})$ denote the state and control sequences from $t_1$ to $t_2$ ($t_1 \leq t_2$), respectively.

The system operates in an environment with $M$ dynamic obstacles with a priori unknown trajectories. Let $Y_t := (Y_{t,1}, ..., Y_{t,M})$ represent the joint obstacle position at time $t$, where $Y_{t,j} \in \mathbb{R}^p$ denotes the position of obstacle $j$ at time $t$. Additionally, the joint obstacle trajectory $Y := (Y_0, ..., Y_T)$ is assumed to be sampled from an unknown probability distribution $\mathcal{D}$, i.e. $Y \sim \mathcal{D}$. The system can observe the joint obstacle states $Y_0, ..., Y_t$, when making the decision at time $t$. We assume the independence between $\mathcal{D}$ and system (1), and the availability of an offline dataset as follows.

**Assumption 3.1.** *For any time $t \geq 0$, the system state $x_{0:t}$ and control $u_{0:t}$ do not change the distribution $\mathcal{D}$.*

**Assumption 3.2.** *We have a calibration dataset $D_{cal} := \{Y^{(1)}, ..., Y^{(N)}\}$, where each of the $N$ joint obstacle trajectories are independently drawn from $\mathcal{D}$, i.e. $Y^{(i)} \sim \mathcal{D}$, $\forall i = 1, ..., N$.*

Assumption 3.1 is typically adopted by default in the literature related to TO Lindemann et al. (2023); Zhu & Alonso-Mora (2019); Hakobyan & Yang (2021) and Assumption 3.2 is not restrictive in practice, e.g. the historical trajectories of obstacles. With Assumptions 3.1 and 3.2, we can conclude that the real joint obstacle trajectory $Y$ and the $N$ available joint obstacle trajectories $Y^{(i)}$ are independent and identically distributed (i.i.d.), and are therefore also exchangeable.

We focus on the TO problem whose objective is to find the sequences $x_{1:T}$ and $u_{0:T-1}$ that minimize the cost function $J(x_{1:T}, u_{0:T-1})$ subject to the dynamics and constraints. The TO is performed in a shrinking-horizon fashion, with the optimization problem at time $t$ formulated as follows.

$$\min_{x_{t+1:T}, u_{t:T-1}} \quad J(x_{t+1:T}, u_{t:T-1}) = l_T(x_T) + \sum_{\tau=t}^{T-1} l_\tau(x_\tau, u_\tau) \tag{2a}$$

$$s.t. \quad x_{\tau+1} = f(x_\tau, u_\tau), \quad \forall \tau = t, ..., T-1 \tag{2b}$$

$$x_\tau \in \mathcal{X}, \quad \forall \tau = t+1, ..., T \tag{2c}$$

$$u_\tau \in \mathcal{U}, \quad \forall \tau = t, ...T-1 \tag{2d}$$

$$\mathbb{P}\left\{\bigcap_{\tau=1}^{T} \{c(x_\tau, Y_\tau) \geq 0\}\right\} \geq 1 - \alpha \tag{2e}$$

where $\mathbb{P}\{X\}$ denotes the probability of event $X$, the constraint function $c := \mathbb{R}^{n_x} \times \mathbb{R}^{Np} \to \mathbb{R}$ is $L$-Lipschitz continuous, which can encode various tasks, such as collision avoidance. Due to the uncertainty of the joint obstacle position $Y_\tau$, we impose the joint chance constraint (2e) with failure probability $\alpha \in (0, 1)$ to ensure that the joint probability of satisfying the constraint over the total mission time is no less than $1 - \alpha$. To ensure the initial feasibility of the TO problem, we assume that the initial state satisfies the constraint, i.e. $c(x_0, Y_0) \geq 0$, with probability 1.

## 3.2 TRAJECTORY PREDICTOR

Recall that system (1) can observe the joint obstacle positions $Y_0, ..., Y_t$ when making decisions at time $t$. By inputting $Y_{0:t}$ into a trajectory prediction algorithm, we can obtain the predictions $\hat{Y}_{t+1|t}, ..., \hat{Y}_{T|t}$ of the future obstacle states $Y_{t+1}, ..., Y_T$. Specifically, a trajectory predictor can be developed by learning a prediction model $g_t : \mathbb{R}^{(t+1)Np} \to \mathbb{R}^{Np}$ from the training dataset $D_{train}$ which is independent of $D_{cal}$. Given an observed joint obstacle trajectory $Y_{0:t}$, the model $g_t(\cdot)$ provides the prediction $\hat{Y}_{t+1|t}$ for the state of the next time $Y_{t+1}$. Then we recursively generate the predictions $\hat{Y}_{t+2|t}, ..., \hat{Y}_{T|t}$ by inputting $\hat{Y}_{1:t+1|t}, ..., \hat{Y}_{T-t-1:T-1|t}$ to the function $g_t(\cdot)$. A specific example of $g_t(\cdot)$ is modeled by Recurrent Neural Network (RNN) which demonstrates significant performance in time series prediction Rudenko et al. (2020). In this paper, we employ Long Short-Term Memory (LSTM) Graves & Graves (2012) to generate the predictions of joint obstacle trajectories. Note that $g_t(\cdot)$ can be any prediction algorithm in our proposed framework.

## 3.3 CONFORMAL PREDICTION

CP is used to obtain prediction regions for predictive models without making any assumptions on the data distribution or the predictive models Vovk et al. (2005); Shafer & Vovk (2008). Here we provide a brief introduction to the theoretical results for CP and refer readers to Angelopoulos & Bates (2021) for a thorough introduction.

Given $N + 1$ exchangeable random variables $R, R^{(1)}, ..., R^{(N)}$, CP aims to find a probabilistic upper bound for $R$ based on $R^{(1)}, ..., R^{(N)}$ such that $R$ is less than this upper bound with high probability. In practice, $R$ represents the test datapoint, while $R^{(1)}, ..., R^{(N)}$ denote the calibration dataset. Formally, the central idea behind CP is summarized in the following lemma.

**Lemma 3.1.** *[Lemma 1 in Tibshirani et al. (2019)] If $R, R^{(1)}, ..., R^{(N)}$ are $N + 1$ exchangeable random variables, then for a failure probability $\alpha \in (0, 1)$, it holds that*

$$\mathbb{P}\left\{R \leq Quantile_{1-\alpha}(R^{(1)}, ..., R^{(N)}, \infty)\right\} \geq 1 - \alpha \tag{3}$$

*where the function $Quantile_{1-\alpha}(R^{(1)}, ..., R^{(N)}, \infty)$ denotes the level $1-\alpha$ quantile of the empirical distribution of the values $R^{(1)}, ..., R^{(N)}, \infty$ as follows.*

$$Quantile_{1-\alpha}(R^{(1)}, ..., R^{(N)}, \infty) = \inf\{z : \mathbb{P}\{Z \leq z\} \geq 1 - \alpha\}, \tag{4a}$$

$$Z \sim \left(\sum_{i=1}^{N} \delta_{R^{(i)}} + \delta_{\infty}\right)/(N + 1) \tag{4b}$$

*where $\delta_{R^{(i)}}$ and $\delta_{\infty}$ denote the Dirac delta function at $R^{(i)}$ and $\infty$, respectively.*

The variable $R$ is usually referred to as the nonconformity score, whose common choice in regression is the prediction error $R := |Y_\tau - \hat{Y}_{\tau|t}|$, where $\hat{Y}_{\tau|t}$ is the prediction of $Y_\tau$.

## 4 END-TO-END CONFORMAL PREDICTION

The challenge in solving the TO problem (2) lies in the computation of the joint probability (2e). Existing literature predominantly employs a sequential way of using CP, i.e. the prediction regions of obstacle positions are first computed based on the failure probability $\alpha$, and then the decision of TO depends one-way on the prediction regions. However, it is important to note that in the shrinking-horizon TO framework, at time $t$ the past decisions $x_{0:t}$ are available and typically contain rich information that can instrumentally assist in refining the prediction regions at subsequent time steps, thereby considerably improving the performance of TO. Therefore, we propose a novel E2E-CP. In particular, E2E-CP not only exploits the feedback information provided by realized trajectories to perform end-to-end adjustments of the prediction regions but also maintains coverage guarantees.

To begin with, the joint chance constraint (2e) can be reformulated as a set of individual chance constraints and a total risk constraint by using Boole's inequality as follows.

$$\mathbb{P}\left\{\bigcap_{\tau=1}^{T}\{c(x_\tau, Y_\tau) \geq 0\}\right\} \geq 1 - \alpha \iff \left\{\begin{array}{l} \mathbb{P}\{c(x_\tau, Y_\tau) \geq 0\} \geq 1 - \alpha_\tau, \ \forall \tau = 1, ..., T \\ \sum_{\tau=1}^{T} \alpha_\tau \leq \alpha \end{array}\right. \tag{5}$$

The risk $\alpha_\tau$ at each time can be initially allocated uniformly at time $t = 0$, i.e. $\alpha_\tau = \alpha/T$, and remains constant throughout the shrinking-horizon TO process, as in Lindemann et al. (2023). However, at time $t$, the system states $x_\tau$ for $\tau \leq t$ are available, which grants us to compute the posterior probability $\beta_\tau = \mathbb{P}\{c(x_\tau, Y_\tau) > 0 | x_\tau\}$ and the permissible risk for future times, which is then used to adjust the prediction regions. Using the information in the realized trajectories, the end-to-end adaptation of the prediction regions tremendously reduces the conservatism of trajectory online while ensuring coverage guarantees. In Subsection 4.1, we present the individual chance constraint reformulation using the prediction regions derived based on a specific risk allocation. In Subsection 4.2, we present a CP-based method for calculating $\beta_\tau$. We reformulate the TO problem in Subsection 4.3. The specific details of the risk allocation are deferred to Section 5.

## 4.1 Constraint reformulation using conformal prediction region

We randomly divide the calibration dataset $D_{cal}$ into two subsets $D_{cal}^1$ and $D_{cal}^2$ with $K$ and $L$ joint obstacle trajectories, respectively, where $K + L = N$. Without loss of generality, we reassign indices to the joint obstacle trajectories as $D_{cal}^1 := \{Y^{(1)}, ..., Y^{(K)}\}$ and $D_{cal}^2 := \{Y^{(K+1)}, ..., Y^{(K+L)}\}$. At time $t$, we can obtain the prediction of the joint obstacle position $\hat{Y}_{\tau|t}$ for all future time $\tau = t + 1, ..., T$ using $g_t(\cdot)$ described in Section 3.2. Similarly, the prediction $\hat{Y}_{\tau|t}^{(i)}$ for each trajectory $Y^{(i)}$ in $D_{cal}^1$ is derived by using the same method. We define the nonconformity score as follows.

$$R_{\tau|t} = \|Y_\tau - \hat{Y}_{\tau|t}\| \qquad R_{\tau|t}^{(i)} = \|Y_\tau^{(i)} - \hat{Y}_{\tau|t}^{(i)}\| \quad \forall i = 1, ..., K \tag{6}$$

Note that $Y_\tau, Y_\tau^{(1)}, ..., Y_\tau^{(K)}$ are exchangeable and the prediction function $g_t(\cdot)$ is trained from $D_{train}$ independent of $D_{cal}^1$. Therefore, given an allocated risk $\alpha_\tau$ for future time $\tau$, the random variables $R_{\tau|t}, R_{\tau|t}^{(1)}, ..., R_{\tau|t}^{(K)}$ are exchangeable and the prediction region with coverage guarantee is derived according to Lemma 3.1 as follows.

$$\mathbb{P}\{\|Y_\tau - \hat{Y}_{\tau|t}\| \leq C_{\tau|t}^{1-\alpha_\tau}\} \geq 1 - \alpha_\tau \tag{7a}$$

$$C_{\tau|t}^{1-\alpha_\tau} = Quantile_{1-\alpha_\tau}(R_{\tau|t}^{(1)}, ..., R_{\tau|t}^{(K)}, \infty) \tag{7b}$$

Based on the $(1 - \alpha_\tau)$-coverage prediction region $\{y : \|y - \hat{Y}_{\tau|t}\| \leq C_{\tau|t}^{1-\alpha_\tau}\}$, the individual chance constraint in (5) can be reformulated as the following lemma proven in Appendix A.1.

**Lemma 4.1.** *If Assumptions 3.1 and 3.2 hold and $c(x_\tau, \hat{Y}_{\tau|t}) \geq LC_{\tau|t}^{1-\alpha_\tau}$ is satisfied where $C_{\tau|t}^{1-\alpha_\tau}$ is calculated by (7b), then the individual chance constraint $\mathbb{P}\{c(x_\tau, Y_\tau) \geq 0\} \geq 1 - \alpha_\tau$ is satisfied.*

## 4.2 Posterior probability conditional on past decisions

At time $t$, the states $x_\tau^*$ for all past time $\tau = 1, ..., t$ are deterministic and available to the trajectory optimizer. We assume that $x_\tau^*$ is the true system state at time $\tau$. Note that $x_\tau^*$ is an feasible solution to the TO problem (2) at time $\tau - 1$ with the reformulated constraints through Lemma 4.1. Therefore, the individual chance constraint $\mathbb{P}\{c(x_\tau^*, Y_\tau) \geq 0\} \geq 1 - \alpha_\tau$ is satisfied at time $\tau - 1$ and will be naturally satisfied for all time $\tau' \geq \tau - 1$. However, the constraint violation probability $\alpha_\tau$ is a priori probability allocated before time $\tau$ that tends to overestimate the violation probability and thus leads to conservative results. Fortunately, the determined $x_\tau^*$ allows us to compute the posterior probability of constraint violation $\beta_\tau$, which, as we theoretically prove, is less than $\alpha_\tau$ with high probability. The risk redundancy between $\alpha_\tau$ and $\beta_\tau$ can be allocated across future times. In this way, the information embedded in $x_\tau^*$ is transmitted back from the decision-making end to the CP end to readjust the prediction region end-to-end and to achieve a trajectory with notably improved performance. To compute $\beta_\tau$ using Lemma 3.1, we propose a novel nonconformity score as follows.

$$S_\tau = c(x_\tau^*, Y_\tau) \qquad S_\tau^{(i)} = c(x_\tau^*, Y_\tau^{(i)}) \quad \forall i = 1, ..., K + L \tag{8}$$

We note that $Y_\tau, Y_\tau^{(1)}, ..., Y_\tau^{(K+L)}$ are exchangeable, and if $x_\tau^*$ is fixed and independent of $Y_\tau, Y_\tau^{(1)}, ..., Y_\tau^{(K+L)}$, the random variables $S_\tau, S_\tau^{(1)}, ..., S_\tau^{(K+L)}$ are also exchangeable. However, as $x_\tau^*$ is derived through the TO problem (2) at time $\tau - 1$, it depends on $D_{cal}^1$ and the random variables $S_\tau, S_\tau^{(1)}, ..., S_\tau^{(K)}$ are no longer exchangeable. Therefore, we only use the subset $D_{cal}^2$, i.e.

$S_\tau^{(K+1)}, ..., S_\tau^{(K+L)}$, to compute $\beta_\tau$. The upper bound of the posterior violation probability $\beta_\tau$ is computed via the following lemma, whose proof is given in Appendix A.2.

**Lemma 4.2.** *Assume that $x_\tau^*$ is the true state of the system at time $\tau$ and Assumption 3.1 holds, then the upper bound of the posterior violation probability at time $\tau$ is as follows.*

$$\mathbb{P}\{c(x_\tau^*, Y_\tau) < 0\} \leq \beta_\tau = \left(1 + \sum_{i=1}^{L} \mathbb{I}\left(S_\tau^{(K+i)} < 0\right)\right)/(1+L) \tag{9}$$

*where $\mathbb{I}(\cdot)$ is the indicator function.*

Some might raise the concern that $\beta_\tau$ could be higher than $\alpha_\tau$, which could result in a more conservative trajectory when using $\beta_\tau$ in subsequent times. However, the following corollary proven in Appendix A.3 restricts the upper bound of the expectation of $\beta_\tau$.

**Corollary 4.1.** *Suppose that $\delta \in (0,1)$ and $K$ is sufficiently large ($K > (-\ln \delta)/(2\alpha_\tau^2)$), we have*

$$\mathbb{P}\left\{\mathbb{E}(\beta_\tau) \leq \left(1 + L\left(\alpha_\tau + \sqrt{-\ln \delta/(2K)}\right)\right)/(1+L)\right\} \geq 1 - \delta \tag{10}$$

*Furthermore if $K, L \to \infty$, then $\mathbb{E}(\beta_\tau) \leq \alpha_\tau$ holds with probability one.*

**Remark 4.1.** *We remark that Corollary 4.1 provides a performance guarantee for the proposed method, i.e. the proposed method performs at least as well as the sequential method Lindemann et al. (2023) with high probability. Furthermore, the experiments in Section 6 demonstrate that the proposed method performs significantly better in practice. This is attributed to the conservatism of the inequality (37) in the proof of Corollary 4.1 (Appendix A.3). Since $c(x_\tau^*, Y_\tau)$ contains the information provided by the function $c$ (e.g. the size and shape of the robot and obstacles) and the system state $x_\tau^*$, it typically occurs that $\mathbb{P}\{c(x_\tau^*, Y_\tau) < 0\} \ll \mathbb{P}\{\|Y_\tau - \hat{Y}_{\tau|t}\| > C_{\tau|t}^{1-\alpha_\tau}\}$ in practice.*

By effectively utilizing $\beta_\tau$, we intelligently adjust the allowable risk and the prediction region for future times to improve the performance of the optimized trajectory online.

### 4.3 OPTIMIZATION PROBLEM REFORMULATION

Thus far, by making use of the joint chance constraint reformulation (5), the individual constraint reformulation in Lemma 4.1 and the posterior probability calculation in Lemma 4.2, the TO problem (2) at time $t$ can be transformed as follows.

$$\min_{x_{t+1:T}, u_{t:T-1}, \alpha_{t+1:T}} \quad J(x_{t+1:T}, u_{t:T-1}) = l_T(x_T) + \sum_{\tau=t}^{T-1} l_\tau(x_\tau, u_\tau) \tag{11a}$$

$$s.t. \quad x_{\tau+1} = f(x_\tau, u_\tau), \qquad \forall \tau = t, ..., T-1 \tag{11b}$$

$$x_\tau \in \mathcal{X}, \qquad \forall \tau = t+1, ..., T \tag{11c}$$

$$u_\tau \in \mathcal{U}, \qquad \forall \tau = t, ...T-1 \tag{11d}$$

$$c(x_\tau, \hat{Y}_{\tau|t}) \geq LC_{\tau|t}^{1-\alpha_\tau}, \quad \forall \tau = t+1, ..., T \tag{11e}$$

$$\alpha_\tau \geq 0, \qquad \forall \tau = t+1, ..., T \tag{11f}$$

$$\sum_{\tau=t+1}^{T} \alpha_\tau \leq \alpha - \sum_{\tau=0}^{t} \beta_\tau \tag{11g}$$

where Constraint (11e) ensures the satisfaction of individual chance constraints (5) for future times $\tau = t+1, ..., T$ through Lemma 4.1, and $C_\tau^{1-\alpha}$ is calculated by (7b). Constraint (11f) is imposed to ensure the non-negativity of $\alpha_\tau$. Constraint (11g) is the most important part for end-to-end adjustments of the prediction region and online performance enhancement of the optimized trajectory. It is derived by replacing $\alpha_\tau$ for past time $\tau = 1, ..., t$ in the total risk constraint (5) with $\beta_\tau$ calculated through Lemma 4.2. The information embedded in past decisions $x_1^*, ..., x_t^*$ influences the future values of $\alpha_{t+1}, ..., \alpha_T$ through the calculation of $\beta_1, ..., \beta_t$ thereby reshaping the prediction region of CP in an end-to-end way. Based on Corollary 4.1 and Remark 4.1, $\beta_\tau$ is highly likely to be less than $\alpha_\tau$ in practice. Consequently, using $\beta_\tau$ grants more risk to be reserved for future times, resulting in much compact prediction regions and tremendously improved optimization performance.

However, it is important to note that $C_{\tau|t}^{1-\alpha_\tau}$ depends on $\alpha_\tau$ and $D_{cal}^1$. Consequently, treating $\alpha_{t+1:T}$ as decision variables alongside $x_{t+1:T}$ and $u_{t:T-1}$ would make the optimization problem (11) computationally demanding to solve for larger values of $T$ and $K$. Therefore, we will present an allocation method for $\alpha_{t+1:T}$ that aligns with the optimization problem (11) in the next section.

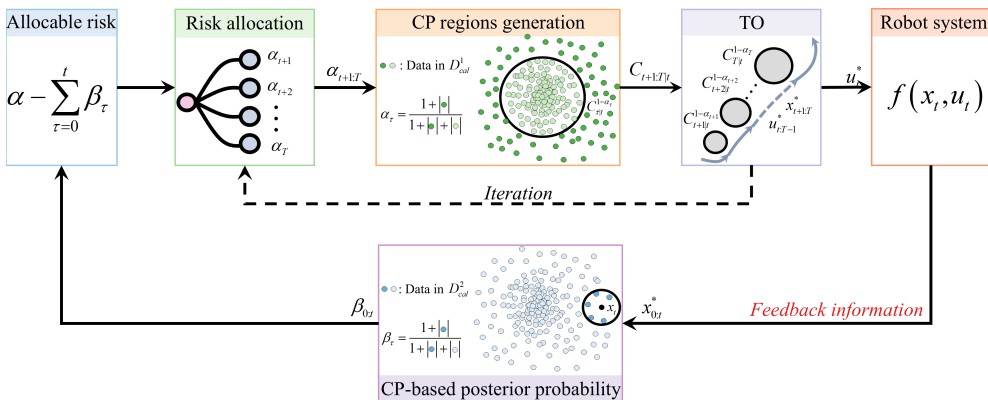

Figure 1: Shrinking-horizon trajectory optimization framework using E2E-CP.

## 5 SHRINKING-HORIZON TRAJECTORY OPTIMIZATION USING E2E-CP

The shrinking-horizon TO framework using E2E-CP is illustrated in Figure 1. The information in $x_{0:t}^*$ guides the end-to-end adjustments of the size of the prediction regions $C_{\tau|t}^{1-\alpha_\tau}$ through posterior probability calculations. Solving the TO problem (11) is divided into two steps: 1) risk allocation and 2) TO with the fixed $\alpha_{t+1:T}$. The TO problem (11) with the fixed $\alpha_{t+1:T}$ is formalized as follows.

$$\min_{x_{t+1:T}, u_{t:T-1}} J(x_{t+1:T}, u_{t:T-1}) \qquad s.t. \quad (11b) - (11e) \tag{12}$$

The problem (12) can be readily solved to obtain $x_{t+1:T}^*$ and $u_{t:T-1}^*$ and only the first system input $u_t^*$ is implemented as the control input. Therefore, as the actual time $t$ progresses, the optimization horizon gradually shrinks. For the risk allocation, a general approach is the Average-based Risk Allocation (ARA), i.e. the allocable risk is evenly allocated across future times at time $t$ below.

$$\alpha_\tau = \left(\alpha - \sum_{\tau=0}^t \beta_\tau\right)/(T-t) \qquad \forall \tau = t+1, ..., T \tag{13}$$

Although the ARA method has the advantage of computational efficiency, the fixed proportion allocation significantly diminishes the flexibility in modifying the prediction regions for future times. Therefore, we extend the IRA proposed in Ono & Williams (2008) to the E2E-CP. To begin with, we refer to the TO problem with a fixed risk allocation $\alpha_{t+1:T}$ (12) and the risk allocation problem as the lower-stage problem and the upper-stage problem, respectively. The system states $x_{t+1:T}^*$ and inputs $u_{t:T-1}^*$, as well as the risk allocation $\alpha_{t+1:T}$ are obtained by iteratively solving the lower and upper-stage problems. We denote the feasible region of the lower-stage problem (12) with $\alpha_{t+1:T}$ as $\mathcal{R}_t(\alpha_{t+1:T})$. The upper-stage problem optimizes $\alpha_{t+1:T}$, formally stated below.

$$\min_{\alpha_{t+1:T}} \quad J^*(\alpha_{t+1:T}) \tag{14a}$$

$$s.t. \quad \alpha_\tau \geq 0, \qquad\qquad \forall \tau = t+1, ..., T \tag{14b}$$

$$\sum_{\tau=t+1}^T \alpha_\tau \leq \alpha - \sum_{\tau=0}^t \beta_\tau \tag{14c}$$

$$\alpha_{t+1:T} \in \{\alpha_{t+1:T} : \exists\, (x_{t+1:t},\ u_{t:T-1}) \in \mathcal{R}_t(\alpha_{t+1:T})\} \tag{14d}$$

where $J^*(\alpha_{t+1:T})$ is the optimal objective function of (12) given $\alpha_{t+1:T}$. If a risk allocation $\alpha_{t+1:T}$ satisfies (14b)-(14d), then we refer to $\alpha_{t+1:T}$ a feasible risk allocation. However, the lower-stage problem (14) is challenging to solve due to the computational complexity arising from its objective function (14a) and Constraint (14c). To solve (14) efficiently, we introduce a descent algorithm, i.e. IRA for E2E-CP. This algorithm is based on the monotonicity of $J^*(\alpha_{t+1:T})$ below, which is theoretically proven in Appendix A.4.

**Lemma 5.1.** *At time t, the following inequalities always hold.*

$$\frac{\partial J^*(\alpha_{t+1:T})}{\partial \alpha_\tau} \leq 0 \quad \forall \tau = t+1, ..., T \tag{15}$$

*where $J^*(\alpha_{t+1:T})$ is defined as the same as in (14a).*

We assume that $\alpha_{t+1:T}^n$ represents the feasible risk allocation obtained after the $n$th iteration at time $t$. IRA aims to obtain a feasible risk allocation $\alpha_{t+1:T}^{n+1}$ in the $(n+1)$th iteration such that $J^*(\alpha_{t+1:T}^{n+1}) \leq J^*(\alpha_{t+1:T}^n)$. In the $(n+1)$th iteration, IRA first solves the lower-stage problem (12) using $\alpha_{t+1:T}^n$ to obtain the optimal solution $x_{t+1:T}^n$ and $u_{t:T-1}^n$. Subsequently, based on $x_{t+1:T}^n$, Constraint (11e) in the lower-stage problem (12) is categorized into active and inactive constraints. The active and inactive constraint sets are formally defined as $\mathcal{I}_{act} := \{\tau : c(x_\tau^n, \hat{Y}_{\tau|t}) = LC_{\tau|t}^{1-\alpha_\tau^n},\ \tau = t+1, ..., T\}$ and $\mathcal{I}_{ina} := \{\tau : \tau \notin \mathcal{I}_{act},\ \tau = t+1, ..., T\}$, respectively. In summary, IRA consists of two steps: 1) tightening the inactive constraints and 2) relaxing the active constraints.

Tightening the inactive constraints is first implemented to construct $\widetilde{\alpha}_{t+1:T}^n$ from $\alpha_{t+1:T}^n$. Specifically, for $\tau \in \mathcal{I}_{act}$, set $\widetilde{\alpha}_\tau^n = \alpha_\tau^n$. Based on the definition of $C_{\tau|t}^{1-\alpha_\tau}$ (7b), $C_{\tau|t}^{1-\alpha_\tau}$ is non-increasing with respect to $\alpha_\tau$ for the fixed $D_{cal}^1$. Thus for $\tau \in \mathcal{I}_{ina}$, we choose $\widetilde{\alpha}_\tau^n \leq \alpha_\tau^n$ so that

$$c(x_\tau^n, \hat{Y}_{\tau|t}) \geq LC_{\tau|t}^{1-\widetilde{\alpha}_\tau^n} \geq LC_{\tau|t}^{1-\alpha_\tau^n} \tag{16}$$

Based on (16), it can be deduced that $(x_{t+1:T}^n, u_{t:T-1}^n) \in \mathcal{R}_t(\widetilde{\alpha}_{t+1:T}^n) \subseteq \mathcal{R}_t(\alpha_{t+1:T}^n)$. Therefore, the optimal solution $(x_{t+1:T}^n, u_{t:T-1}^n)$ for $\alpha_{t+1:T}^n$ is also the optimal solution for $\widetilde{\alpha}_{t+1:T}^n$, and thus $J^*(\alpha_{t+1:T}^n) = J^*(\widetilde{\alpha}_{t+1:T}^n)$. Finally, it is straightforward to show that $\widetilde{\alpha}_{t+1:T}^n$ is a feasible risk allocation, because (i) (14b) follows from (16) and the fact that when $\alpha_\tau \to 0$, $C_{\tau|t}^{1-\alpha_\tau} \to \infty$; (ii) (14c) is satisfied since $\sum_{\tau=t+1}^T \widetilde{\alpha}_\tau^n \leq \sum_{\tau=t+1}^T \alpha_\tau^n \leq \alpha - \sum_{\tau=1}^t \beta_\tau$; (iii) (14d) is satisfied because $(x_{t+1:T}^n, u_{t:T-1}^n)$ is feasible for $\widetilde{\alpha}_{t+1:T}^n$. The specific construction of $\widetilde{\alpha}_\tau^n$ is as follows.

$$\widetilde{\alpha}_\tau^n = \begin{cases} \alpha_\tau^n, & \tau \in \mathcal{I}_{act} \\ (1-\eta)\alpha_\tau^n + \eta\underline{\alpha}_\tau^n, & \tau \in \mathcal{I}_{ina} \end{cases} \tag{17}$$

where $\eta \in (0,1)$ is the step size and $\underline{\alpha}_\tau^n$ is the lower bound of $\widetilde{\alpha}_\tau^n$, $\forall \tau \in \mathcal{I}_{ina}$ calculated as in Lemma 5.2, which is proven in Appendix A.5.

**Lemma 5.2.** *Assume that $x_{t+1:T}^n$ is feasible for the problem (12) with $\alpha_{t+1:T}^n$ and $\alpha_{t+1:T}^n < 1$. For $\tau \in \mathcal{I}_{ina}$, the lower bound of $\widetilde{\alpha}_\tau^n$ while satisfying (16) is as follows.*

$$\underline{\alpha}_\tau^n = \left(1 + \sum_{i=1}^K \mathbb{I}\left(c(x_\tau^n, \hat{Y}_{\tau|t}) < LR_{\tau|t}^{(i)}\right)\right)/(1+K) \tag{18}$$

*Furthermore, it is deterministic that $\underline{\alpha}_\tau^n \leq \alpha_\tau^n$.*

Then $\alpha_{t+1:T}^{n+1}$ is constructed from $\widetilde{\alpha}_{t+1:T}^n$ to relax the active constraints as follows.

$$\alpha_\tau^{n+1} = \begin{cases} \widetilde{\alpha}_\tau^n + \left(\alpha - \sum_{\tau=1}^t \beta_\tau - \sum_{\tau=t+1}^T \widetilde{\alpha}_\tau^n\right)/N_{act}, & \tau \in \mathcal{I}_{act} \\ \widetilde{\alpha}_\tau^n, & \tau \in \mathcal{I}_{ina} \end{cases}\ ; \tag{19}$$

where $N_{act}$ represents the number of elements in the set $\mathcal{I}_{act}$. It can be easily verified that $\alpha_{t+1:T}^{n+1}$ satisfies (14b)-(14d), and thus $\alpha_{t+1:T}^{n+1}$ is a feasible risk allocation. Note that $\alpha_\tau^{n+1} \geq \widetilde{\alpha}_\tau^n$ since $\widetilde{\alpha}_\tau^n$ satisfies (14c). Therefore, the following inequality is obtained by implying Lemma 5.1.

$$J^*(\alpha_{t+1:T}^{n+1}) \leq J^*(\widetilde{\alpha}_{t+1:T}^n) = J^*(\alpha_{t+1:T}^n) \tag{20}$$

By recursively constructing $\alpha_{t+1:T}^1, ..., \alpha_{t+1:T}^n$ in this manner, $J^*$ monotonically decreases. The algorithm of E2E-CP using IRA at time $t$ is delineated in Algorithm 1 (in Appendix B). The convergence of Algorithm 1 is provided in the following theorem proven in Appendix A.6.

**Theorem 5.1.** *Assume that $x_{t+1:T}^0, u_{t:T-1}^0$ are feasible in problem (12) with risk allocation $\alpha_{t+1:T}^0$. If the sets $\mathcal{X}$, $\mathcal{U}$ are bounded and the objective function $J(x_{t+1:T}, u_{t:T-1})$ is continuous, then the sequence of the optimal objective value $\{J^*(\alpha_{t+1:T}^n)\}_{n\in\mathbb{N}}$ converges to a finite limit.*

**Remark 5.1.** *One may notice that the calculation of $\beta_\tau$ in Lemma 4.2 is similar to the computation of $\widetilde{\alpha}_\tau^n$ in Lemma 5.2. This observation is correct. The key difference between the two lies in that the former utilizes the dataset $D_{cal}^2$ independent with $D_{cal}^1$ to achieve the coverage guarantee for $\beta_\tau$. By contrast, as a step in solving (11), the latter does not need to consider the coverage guarantee and thus directly uses $D_{cal}^1$. The use of different datasets results in the former providing probabilistic guarantee (Corollary 4.1), while the latter achieves deterministic guarantee ($\underline{\alpha}_\tau^n \leq \alpha_\tau^n$).*

## 6 EXPERIMENTS

We conduct simulations to demonstrate the effectiveness of the E2E-CP for TO. In particular, we conduct 1,000 Monte Carlo experiments on a kinematic vehicle model Pepy et al. (2006), a 3D linear quadrotor model Mistler et al. (2001), and a dynamic bicycle model Hakobyan & Yang (2021)[1]. We present and analyze experimental results, and the experiment details and comprehensive results can be found in Appendix C. The following TO methods are analyzed in all benchmark experiments[2].

- Sequential CP: Computation of the CP region and TO is performed sequentially, without end-to-end adjustments to the region after decision-making.
- E2E-CP with ARA: The method based on E2E-CP using average risk allocation.
- E2E-CP with IRA: The method based on E2E-CP using iterative risk allocation.

Figure 2 shows the simulation results from one of the 1,000 independent simulations using the 2D vehicle model. At $t = 0$, the vehicle performs the first TO using different methods. For E2E-CP with IRA, IRA allows for flexible allocation of the risks across future times. Therefore, by assigning more risk to the time $\tau = 9$, which leads to a compact prediction region, a trajectory passing between Obstacles 2 and 3 is obtained. However, with the fixed risk allocation at $t = 0$, Sequential CP and E2E-CP with ARA can only optimize the trajectory based on fixed prediction regions. Consequently, they can only navigate around to pass between Obstacles 1 and 2. Note that at $t = 0$, no deterministic vehicle position is available for posterior probability calculation. Thus E2E-CP with ARA degenerates into Sequential CP, resulting in both methods obtaining essentially the same trajectory. As time progresses, more and more vehicle positions become available. For E2E-CP with ARA, $\beta_{1:3}$ can be computed at $t = 3$ and is with high probability less than $\alpha_{1:3}$, as outlined in Corollary 4.1. The reduction from $\alpha_{1:3}$ to $\beta_{1:3}$ leads to an increased allowable risk for future times, corresponding to a narrowing in the prediction regions. As a result, compared with Sequential CP, E2E-CP with ARA generates a less conservative trajectory. Similarly, E2E-CP with IRA also leverages $\beta_{1:3}$ to increase the total allocable risk, thereby further enhancing the flexibility in allocating risks for future times. As illustrated in Figure 2, the trajectory obtained by E2E-CP with IRA at $t = 3$ exhibits reduced conservativeness compared with the trajectory obtained at $t = 0$.

Table 1 summarizes the average cost, average computation time, and collision avoidance rate of 1,000 simulations with different methods. The complete tables can be found in Appendix C. As shown in Table 1, the E2E-CP with ARA noticeably reduces the cost by an average of 11.26%

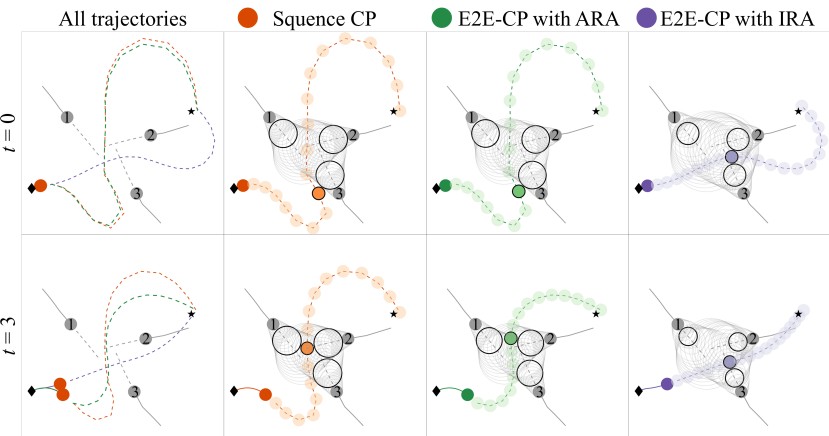

Figure 2: Trajectories of the vehicle with different TO methods. (Numbers on the circles denote the indices of obstacles. The diamond and pentagon symbols represent the initial and target points of the vehicle, respectively. The translucent circles represent the planned positions of the vehicle and the prediction regions for further time. In particular, the colored and transparent circles with black edges denote the planned positions and the prediction regions for $\tau = 9$, respectively.)

---

[1] The interior-point method-based solver IPOPT (v3.12.9) was used to solve the TO problem (11).

[2] Source code will be made available upon acceptance of the paper.

Table 1: Average cost, computation time, and collision avoidance rate using the vehicle model with different methods. The total risk tolerance is set to $\alpha = 0.2$.

|  | Sequential CP | E2E-CP with ARA | E2E-CP with IRA |
|---|---|---|---|
| Average cost | 17.05 | 15.13 | 2.89 |
| Average computation time (s) | 0.076 | 0.078 | 0.131 |
| Collision avoidance rate | 88.4% | 89.1% | 91.2% |

compared with Sequential CP thanks to the feedback information of posterior probabilities, with a negligible additional computational burden. Furthermore, by flexibly allocating the additional allowable risk provided by posterior probabilities, E2E-CP with IRA achieves an 83.05% reduction in average cost compared with Sequential CP. However, since IRA needs to solve the TO problem (12) iteratively, the average computation time increases significantly. Additionally, thanks to the coverage guarantee for E2E-CP provided by Lemma 4.2, both E2E-CP with ARA and IRA achieve a collision avoidance rate that exceeds the required threshold (80%). Thus ARA and IRA can be freely chosen according to the computational capability, real-time requirement, and trajectory performance.

To investigate the impact of using prior versus posterior probabilities on the prediction regions, we collect the prediction region radius for time $t$, denoted as $C_{20|t}$, using the vehicle model with different methods across 1,000 simulations, as illustrated in Figure 3. It can be observed that $C_{20|t}$ decreases as $t$ increases, which is reasonable since the error of the trajectory predictor diminishes as $t$ approaches $\tau$. Note that since Sequential CP only uses prior probabilities to compute $C_{20|t}$ throughout the entire planning process, which depends solely on $D_{cal}$, $C_{20|t}$ remains constant for a fixed $t$ across the 1,000 simulations. By contrast, for E2E-CP with ARA, $C_{20|t}$ also depends on the actual obstacle positions and past decisions due to the use of the posterior probabilities, which leads to the variability of $C_{20|t}$ across 1,000 simulations. The distribution of $C_{20|9}$ is illustrated in the right panel of Figure 3. It can be seen that $C_{20|t}$ computed by E2E-CP with ARA is typically smaller than that computed by Sequential CP. As $t$ increases, more posterior probabilities can be used, leading to a growing gap between the $C_{20|t}$ calculated by the two methods, which corroborates Corollary 4.1. The details about the prediction region radius for different $t$ and $\tau$ are provided in Appendix F.

Although our experiments are conducted under exchangeability provided by Assumptions 3.1 and 3.2, we have empirically demonstrated the proposed method exhibits a certain degree of robustness to moderate distribution shifts and can maintain safety and high performance in realistic scenarios (beyond Assumption 3.1). Detailed experiments can be found in Appendix D and Appendix E.

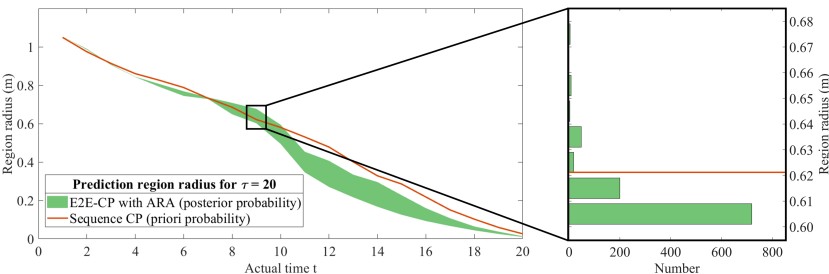

Figure 3: Left: prediction region radius for $\tau = 20$ at each time $t$ ($C_{20|t}$) using the vehicle model with different methods across 1,000 simulations. Right: distributions of $C_{20|9}$.

## 7 CONCLUSION AND LIMITATIONS

In this paper, we proposed an E2E-CP framework for shrinking-horizon TO with a joint risk constraint over the entire mission time in uncertain environments. This method enables the feedback of the information in the realized trajectory from the decision-making end to the CP end, guiding the end-to-end adjustments of the prediction regions. The proposed end-to-end adjustment rule balances both performance and safety, offering provable performance and coverage guarantees. Furthermore, the proposed E2E-CP is not limited to TO, it can be applied to any safety-critical decision-making.

The proposed E2E-CP has two limitations: the requirement for a sufficient calibration dataset size and the reliance of the theoretical guarantees in this paper on data exchangeability. We provide a detailed discussion and several potential ways to address these limitations in Appendix G.

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

# A  PROOFS

## A.1  PROOF OF LEMMA 4.1

According to Assumptions 3.1 and 3.2 as well as the calculation of $C_{\tau|t}^{1-\alpha_\tau}$ (7b), the $(1 - \alpha_\tau)$-coverage guarantee of the prediction (7a) is obtained through Lemma 3.1. Note that the function $c$ is $L$-Lipschitz continuous, the following inequality is obtained.

$$\|c(x_\tau, Y_\tau) - c(x_\tau, \hat{Y}_{\tau|t})\| \le L\|Y_\tau - \hat{Y}_{\tau|t}\| \implies c(x_\tau, Y_\tau) \ge c(x_\tau, \hat{Y}_{\tau|t}) - L\|Y_\tau - \hat{Y}_{\tau|t}\| \quad (21)$$

If the constraint $c(x_\tau, \hat{Y}_{\tau|t}) \ge L C_{\tau|t}^{1-\alpha_\tau}$ is satisfied, we have the following inequality.

$$c(x_\tau, Y_\tau) \ge L(C_{\tau|t}^{1-\alpha} - \|Y_\tau - \hat{Y}_{\tau|t}\|) \quad (22)$$

According to the $(1 - \alpha_\tau)$-coverage guarantee (7a) $\mathbb{P}\{C_{\tau|t}^{1-\alpha} - \|Y_\tau - \hat{Y}_{\tau|t}\| \ge 0\} \ge 1 - \alpha_\tau$, the lemma is proven. □

## A.2  PROOF OF LEMMA 4.2

Based on Assumption 3.1, the random variables $Y_\tau, Y_\tau^{(K+1)}, ..., Y_\tau^{(K+L)}$ are exchangeable. Note that $x_\tau^*$ is the true state of the system at time $\tau$, thus $x_\tau^*$ is fixed and independent of $Y_\tau, Y_\tau^{(K+1)}, ..., Y_\tau^{(K+L)}$. Therefore, the random variables $S_\tau, S_\tau^{(K+1)}, ..., S_\tau^{(K+L)}$ are exchangeable.

Without loss of generality, we assume that the dataset $\{-S_\tau^{(K+i)} : i = 1, ..., L\}$ are sorted in non-decreasing order. We first assume that $-S_\tau^{(K+1)} \le 0$, and then we define the maximum index $\ell$ that makes $-S_\tau^{(K+\ell)} \le 0$ hold as follows.

$$\ell = \max_{l=1,...,L} l \\ s.t. \quad -S_\tau^{(K+\ell)} \le 0 \quad (23)$$

Then the posterior satisfaction probability can be computed below.

$$\mathbb{P}\{c(x_\tau^*, Y_\tau) \ge 0\} = \mathbb{P}\{-S_\tau \le 0\} \ge \mathbb{P}\{-S_\tau \le -S_\tau^{(K+\ell)}\} \quad (24)$$

It is assumed that there are $t$ terms in $\{-S_\tau^{(K+i)} : i = 1, ..., L\}$ identical to $-S_\tau^{(K+\ell)}$, i.e.

$$-S_\tau^{(K+\ell-t)} < -S_\tau^{(K+\ell-t+1)} = ... = -S_\tau^{(K+\ell)} \le 0 < -S_\tau^{(K+\ell+1)} \quad (25)$$

Then $-S_\tau^{(K+\ell)}$ can be equivalently reformulated as follows.

$$-S_\tau^{(K+\ell)} = Quantile_\beta(-S_\tau^{(K+1)}, ..., -S_\tau^{(K+L)}, \infty), \quad \forall \beta \in \left(\frac{\ell-t}{1+L}, \frac{\ell}{1+L}\right] \quad (26)$$

Combining (24) and (26) we have

$$\mathbb{P}\{c(x_\tau^*, Y_\tau) \ge 0\} \ge \mathbb{P}\{-S_\tau \le Quantile_\beta(-S_\tau^{(K+1)}, ..., -S_\tau^{(K+L)}, \infty)\} \quad (27)$$

Note that the random variables $S_\tau, S_\tau^{(K+1)}, ..., S_\tau^{(K+L)}$ are exchangeable and $\beta \in \left(\frac{\ell-t}{1+L}, \frac{\ell}{1+L}\right] \subset (0, 1)$, and thus we can apply Lemma 3.1 and obtain

$$\mathbb{P}\{c(x_\tau^*, Y_\tau) \ge 0\} \ge \beta, \quad \forall \beta \in \left(\frac{\ell-t}{1+L}, \frac{\ell}{1+L}\right] \quad (28)$$

Therefore, the upper bound of the posterior violation probability can be computed by

$$\mathbb{P}\{c(x_\tau^*, Y_\tau) < 0\} \le 1 - \beta, \quad \forall \beta \in \left(\frac{\ell-t}{1+L}, \frac{\ell}{1+L}\right] \quad (29)$$

To minimize this upper bound, we take the maximum value of $\beta$ and (29) becomes (30).

$$\mathbb{P}\{c(x_\tau^*, Y_\tau) < 0\} \le 1 - \frac{\ell}{1+L} \quad (30)$$

According to the definition of $\ell$ (23), we can compute $\ell$ as follows.

$$\ell = \sum_{i=1}^{L} \mathbb{I}\left(S_\tau^{(K+i)} \geq 0\right) = L - \sum_{i=1}^{L} \mathbb{I}\left(S_\tau^{(K+i)} < 0\right) \tag{31}$$

Combining (30) and (31), we have

$$\mathbb{P}\{c(x_\tau^*, Y_\tau) < 0\} \leq \frac{1 + \sum_{i=1}^{L} \mathbb{I}\left(S_\tau^{(K+i)} < 0\right)}{1 + L} \tag{32}$$

Finally, we consider the scenario in which $-S_\tau^{(K+1)} > 0$, which means $S_\tau^{(K+i)} < 0 \, \forall i = 1, ..., L$. Then the inequality (32) is simplified as follows.

$$\mathbb{P}\{c(x_\tau^*, Y_\tau) < 0\} \leq 1 \tag{33}$$

which is always true. Thus, the Lemma is proven. $\qquad \square$

## A.3 Proof of Corollary 4.1

Taking expectations on both sides of Equation (9), we can obtain

$$\mathbb{E}(\beta_\tau) = \frac{1 + L\mathbb{P}\{S_\tau^{(K+i)} < 0\}}{1 + L} = \frac{1 + L\mathbb{P}\{S_\tau < 0\}}{1 + L} = \frac{1 + L\mathbb{P}\{c(x_\tau^*, Y_\tau) < 0\}}{1 + L} \tag{34}$$

The second equality holds because $x_\tau^*$ is fixed and independent of $Y_\tau, Y_\tau^{(K+1)}, ..., Y_\tau^{(K+L)}$, and since $Y_\tau, Y_\tau^{(K+1)}, ..., Y_\tau^{(K+L)}$ are i.i.d., $S_\tau, S_\tau^{K+1}, ..., S_\tau^{K+L}$ are also i.i.d.. Note that the function $c$ is $L$-Lipschitz continuous and $x_\tau^*$ is a feasible solution of problem (2) with the reformulated constraint through Lemma 4.1, and the following inequality can be derived in the same manner as inequalities (21) and (22) in the Proof of Lemma 4.1 (Appendix A.1).

$$c(x_\tau^*, Y_\tau) \geq L(C_{\tau|t}^{1-\alpha_\tau} - \|Y_\tau - \hat{Y}_{\tau|t}\|) \tag{35}$$

Based on (35), we can obtain

$$c(x_\tau^*, Y_\tau) < 0 \Rightarrow \|Y_\tau - \hat{Y}_{\tau|t}\| > C_{\tau|t}^{1-\alpha_\tau} \tag{36}$$

And the following inequality is derived.

$$\mathbb{P}\{c(x_\tau^*, Y_\tau) < 0\} \leq \mathbb{P}\{\|Y_\tau - \hat{Y}_{\tau|t}\| > C_{\tau|t}^{1-\alpha_\tau}\} \tag{37}$$

Combining (34) and (37), we have

$$\mathbb{E}(\beta_\tau) \leq \frac{1 + L\mathbb{P}\{\|Y_\tau - \hat{Y}_{\tau|t}\| > C_{\tau|t}^{1-\alpha_\tau}\}}{1 + L} \tag{38}$$

For $\alpha_\tau, \delta \in (0, 1)$ and $K > (-\ln \delta)/(2\alpha_\tau^2)$, we can apply [Vovk (2012), Proposition 2a] so that

$$\mathbb{P}\left\{\mathbb{P}\left\{\|Y_\tau - \hat{Y}_{\tau|t}\| \leq C_{\tau|t}^{1-\alpha_\tau}\right\} \geq 1 - \left(\alpha_\tau + \sqrt{-\ln \delta/(2K)}\right)\right\} \geq 1 - \delta \tag{39}$$

which can be equivalently transformed into the following expression.

$$\mathbb{P}\left\{\mathbb{P}\left\{\|Y_\tau - \hat{Y}_{\tau|t}\| > C_{\tau|t}^{1-\alpha_\tau}\right\} \leq \alpha_\tau + \sqrt{-\ln \delta/(2K)}\right\} \geq 1 - \delta \tag{40}$$

Combining (38) and (40), we can finally obtain the inequality (10).

When $K, L \to \infty$, we can further assume that $L \geq 1/\delta$ and $K \geq \max\{(-\ln \delta)/(2\alpha_\tau^2), 1/\delta\}$. Note that for a fixed $\alpha_\tau$, we can always find a small enough positive $\delta$ such that $\alpha_\tau + \sqrt{(-\ln \delta)/(2K)} < \alpha_\tau + \sqrt{(-\delta \ln \delta)/2} < 1$. Therefore for a small enough positive $\delta$ we have

$$\mathbb{P}\left\{\mathbb{E}(\beta_\tau) \leq \frac{\delta + \alpha_\tau + \sqrt{-\delta \ln \delta/2}}{\delta + 1}\right\}$$

$$\geq \mathbb{P}\left\{\mathbb{E}(\beta_\tau) \leq \frac{1 + L\left(\alpha_\tau + \sqrt{-\ln \delta/(2K)}\right)}{1 + L}\right\} \geq 1 - \delta \tag{41}$$

Let $\delta \to 0^+$, we finally obtain that $\mathbb{E}(\beta_\tau) \leq \alpha_\tau$ holds with probability one. $\qquad \square$

## A.4 Proof of Lemma 5.1

Let $\alpha_{t+1:T}^1$ and $\alpha_{t+1:T}^2$ be two risk allocations at time $t$. Based on the definition of $C_{\tau|t}^{1-\alpha_\tau}$ (7b), $C_{\tau|t}^{1-\alpha_\tau}$ is non-increasing with respect to $\alpha_\tau$ for fixed $D_{cal}^1$. Therefore, if $\alpha_\tau^1 \leq \alpha_\tau^2$, $\forall \tau = t+1, ..., T$, then $C_{\tau|t}^{1-\alpha_\tau^1} \geq C_{\tau|t}^{1-\alpha_\tau^2}$ which further leads to $\mathcal{R}_t(\alpha_{t+1:T}^1) \subseteq \mathcal{R}_t(\alpha_{t+1:T}^2)$. Since $J^*(\alpha_{t+1:T})$ is the minimum of the objective problem (12) with the feasible region $\mathcal{R}_t(\alpha_{t+1:T})$, $J^*(\alpha_{t+1:T}^1) \geq J^*(\alpha_{t+1:T}^2)$ can be obtained and the lemma is proven. $\qquad\square$

## A.5 Proof of Lemma 5.2

The computation of the lower bound is analogous to the calculation of $\beta_\tau$ in Lemma 4.1, except that Lemma 3.1 is not required to obtain coverage guarantees. Therefore, the computation is based on $D_{cal}^1$. Without loss of generality, we assume that the dataset $\{R_{\tau|t}^{(i)} : i = 1, ..., K\}$ is sorted in non-decreasing order. Note that $x_{t+1:T}^n$ is feasible for the problem (12) with $\alpha_{t+1:T}^n$ and $\tau \in \mathcal{I}_{ina}$, the inequality $c(x_\tau^n, \hat{Y}_{\tau|t}) > LC_{\tau|t}^{1-\alpha_\tau^n} = LQuantile_{1-\alpha_\tau^n}(R_{\tau|t}^{(1)}, ..., R_{\tau|t}^{(K)}, \infty)$ holds true. Since $\alpha_\tau^n < 1$, it follows that $c(x_\tau^n, \hat{Y}_{\tau|t}) > R_{\tau|t}^{(1)}$. Therefore, we define the maximum index $\mathcal{K}$ that makes $c(x_\tau^n, \hat{Y}_{\tau|t}) \geq LR_{\tau|t}^{(\mathcal{K})}$ hold as follows.

$$\mathcal{K} = \max_{k=1,...,K} \quad k$$
$$s.t. \quad c(x_\tau^n, \hat{Y}_{\tau|t}) \geq LR_{\tau|t}^{(k)} \tag{42}$$

It is assumed that there are $t$ terms in $\{R_{\tau|t}^{(i)} : i = 1, ..., K\}$ identical to $R_{\tau|t}^{\mathcal{K}}$, and thus we can obtain

$$R_{\tau|t}^{(\mathcal{K}-t)} < R_{\tau|t}^{(\mathcal{K}-t+1)} = ... = R_{\tau|t}^{(\mathcal{K})} \leq c(x_\tau^n, \hat{Y}_{\tau|t})/L < R_{\tau|t}^{(\mathcal{K}+1)} \tag{43}$$

We aim to determine the maximum value of $C_{\tau|t}^{1-\widetilde{\alpha}_\tau^n}$ (the minimum value of $\widetilde{\alpha}_\tau^n$) while satisfying $C_{\tau|t}^{1-\widetilde{\alpha}_\tau^n} \leq c(x_\tau^n, \hat{Y}_{\tau|t})/L$, which is equivalent to $C_{\tau|t}^{1-\widetilde{\alpha}_\tau^n} \leq R_{\tau|t}^{\mathcal{K}}$ because $C_{\tau|t}^{1-\widetilde{\alpha}_\tau^n}$ can only take values at a finite number of discrete points $R_{\tau|t}^{(1)}, ..., R_{\tau|t}^{(K)}, \infty$. Furthermore, $R_{\tau|t}^{\mathcal{K}}$ can be equivalently reformulated as follows.

$$R_{\tau|t}^{\mathcal{K}} = Quantile_\beta(R_{\tau|t}^{(1)}, ..., R_{\tau|t}^{(K)}, \infty) = C_{\tau|t}^\beta, \quad \forall \beta \in \left( \frac{\mathcal{K}-t}{1+K}, \frac{\mathcal{K}}{1+K} \right] \tag{44}$$

Therefore, the constraint $C_{\tau|t}^{1-\widetilde{\alpha}_\tau^n} \leq R_{\tau|t}^{\mathcal{K}}$ is equivalent to the following expression.

$$C_{\tau|t}^{1-\widetilde{\alpha}_\tau^n} \leq C_{\tau|t}^\beta, \quad \exists \beta \in \left( \frac{\mathcal{K}-t}{1+K}, \frac{\mathcal{K}}{1+K} \right] \tag{45}$$

Note that $C_{\tau|t}^\beta$ is non-decreasing with respect to $\beta$ for fixed $D_{cal}^1$. Constraint (45) is further reformulated as follows.

$$\widetilde{\alpha}_\tau^n \geq 1 - \frac{\mathcal{K}}{1+K} \tag{46}$$

According to the definition of $\mathcal{K}$ (42), we can compute $\mathcal{K}$ in (47).

$$\mathcal{K} = \sum_{i=1}^K \mathbb{I}\left( c(x_\tau^n, \hat{Y}_{\tau|t}) \geq LR_{\tau|t}^{(i)} \right) = K - \sum_{i=1}^K \mathbb{I}\left( c(x_\tau^n, \hat{Y}_{\tau|t}) < LR_{\tau|t}^{(i)} \right) \tag{47}$$

Combining (46) and (47), the lower bound of $\widetilde{\alpha}_\tau^n$ is computed as follows.

$$\underline{\alpha}_\tau^n = \frac{1 + \sum_{i=1}^K \mathbb{I}\left( c(x_\tau^n, \hat{Y}_{\tau|t}) < LR_{\tau|t}^{(i)} \right)}{1+K} \tag{48}$$

We note that $\underline{\alpha}_\tau^n$ is the lower bound of $\widetilde{\alpha}_\tau^n$ that ensures the constraint $c(x_\tau^n, \hat{Y}_{\tau|t}) \geq LC_{\tau|t}^{1-\widetilde{\alpha}_\tau^n}$. Furthermore, since $x_{t+1:T}^n$ is feasible for the problem (12) with $\alpha_{t+1:T}^n$, the constraint $c(x_\tau^n, \hat{Y}_{\tau|t}) \geq LC_{\tau|t}^{1-\alpha_\tau^n}$ is satisfied. Therefore, $\underline{\alpha}_\tau^n \leq \alpha_\tau^n$ is naturally obtained. Thus the Lemma is proven. $\qquad\square$

### A.6 Proof of Theorem 5.1

The proof adapts elements of the proof from Zymler et al. (2013). If $x_{t+1:T}^0, u_{t:T-1}^0$ is a feasible solution for the risk allocation $\alpha_{t+1:T}^0$, the update law of $\alpha_{t+1:T}$ guarantees that the sequence of the optimal objective values $\{J^*(\alpha_{t+1:T}^n)\}_{n \in \mathbb{N}}$ is monotonically decreasing, as previously mentioned. Since the sets $\mathcal{X}$ and $\mathcal{U}$ are bounded, $x_{t+1:T}$ and $u_{t:T-1}$ are bounded. Because the objective function $J(x_{t+1:T}, u_{t:T-1})$ is continuous, the boundedness of $x_{t+1:T}$, $u_{t:T-1}$ and the monotonicity of the optimal objective value sequence imply that $\{J^*(\alpha_{t+1:T}^n)\}_{n \in \mathbb{N}}$ converges to a finite limit. $\qquad\square$

## B Algorithm

The algorithm of E2E-CP using IRA at time $t$ is delineated as follows. Note that at time $t = 0$, the input parameter $\alpha_{0:T}$ is initialized as $\alpha_0 = 0$, $\alpha_{1:T} = \alpha/T$, $\beta_{0:t-1}$ is omitted, and the posterior probability calculation in Line 3 is replaced by the assignment $\beta_0 = 0$. $\epsilon$ is a given small tolerance.

---

**Algorithm 1:** E2E-CP using IRA at time $t$

---

**Input:** $\alpha$, $\alpha_{t:T}$, $\beta_{0:t-1}$, $\epsilon$, $\eta$, $D_{cal}^1$, $D_{cal}^2$
1   Observe the system state $x_t$ and joint obstacle states $Y_t$ ;
2   $\hat{Y}_{t+1|t}, ..., \hat{Y}_{T|t} \leftarrow$ Trajectory prediction using LSTMs based on $Y_0, ..., Y_t$ ;
3   $\beta_t \leftarrow$ Posterior probability calculation (9) ;       // Using $x_t$ and $D_{cal}^2$
4   $J^*(\alpha_{t+1:T}^{-1}) \leftarrow \infty$, $\alpha_{t+1:T}^0 \leftarrow \alpha_{t+1:T}$, $n \leftarrow 0$ ;     // Initialization of IRA
5   **repeat**
6     $J^*(\alpha_{t+1:T}^n)$, $x_{t+1:T}^n$, $u_{t:T-1}^n \leftarrow$ Solve the lower-stage problem (12) with $\alpha_{t+1:T}^n$ ;
7     $\mathcal{I}_{act}$, $\mathcal{I}_{ina}$, $N_{act} \leftarrow$ Identification of active and inactive constraints ;
8     $\widetilde{\alpha}_{t+1:T}^n \leftarrow$ Transitional risk allocation calculation (17) ;
9     $\alpha_{t+1:T}^{n+1} \leftarrow$ New risk allocation calculation (19) ;
10   $n \leftarrow n + 1$ ;
11 **until** $|J^*(\alpha_{t+1:T}^{n-1}) - J^*(\alpha_{t+1:T}^{n-2})| < \epsilon$;
    **Output:** $\beta_{0:t}$, $u_{t:T-1}^{n-1}$, $\alpha_{t+1:T} = \alpha_{t+1:T}^{n-1}$,

---

## C Experiment details and additional results

### C.1 Simulation for a kinematic vehicle model

We examine the kinematic vehicle model Pepy et al. (2006) with the following nonlinear dynamics.

$$\begin{bmatrix} p_{x,t+1} \\ p_{y,t+1} \\ \theta_{t+1} \\ v_{t+1} \end{bmatrix} = \begin{bmatrix} p_{x,t} + \Delta v_t \cos \theta_t \\ p_{y,t} + \Delta v_t \sin \theta_t \\ \theta_t + \Delta \frac{v_t}{l} \tan \phi_t \\ v_t + \Delta a_t \end{bmatrix} \tag{49}$$

where $p_t := (p_{x,t}, p_{y,t})$, $\theta_t$, $v_t$ are the position, orientation, and velocity of the vehicle, respectively. $l := 0.2$ is the length, and $\Delta = 0.125$ is the sampling time. The system inputs are the steering angle $\phi_t \in [-\pi/6, \pi/6]$ and the acceleration $a_t \in [-5, 5]$. The total time is set to $T = 20$. The objective is to reach the vicinity of the target point while avoiding collisions with obstacles. Formally, the objective function is defined as $J = \sum_{\tau=t}^{T-1} 100\phi_\tau^2 + a_\tau^2$ to minimize energy consumption and the constraint $\|p_T - p_{tar}\|_2 \leq 0.2$ is incorporated into (11) to ensure the vehicle reaches the target point, where $p_{tar}$ is the target point. The constraint function for collision avoidance is as follows.

$$c(p_\tau, Y_\tau) = \min_{j=1,...,M} \|p_\tau, Y_{\tau,j}\|_2 - r_r - r_o - r_s \tag{50}$$

where $r_r$ and $r_o$ are the inflation radius of the vehicle and obstacle, respectively. $r_s$ is the safety margin. Similar to Lindemann et al. (2023), we consider $M = 3$ obstacles, with their trajectories generated by TrajNet++ Kothari et al. (2021) using the ORCA simulator Van den Berg et al. (2008).

We generate 13,000 joint obstacle trajectories and randomly divide them into training $D_{train}$, calibration $D_{cal}$, and test $D_{test}$ datasets with the set sizes 2,000, 10,000, and 1,000, respectively. We train an LSTM Alahi et al. (2016) using $D_{train}$ as the trajectory predictor. For the proposed E2E-CP, $D_{cal}$ is further divided into $D_{cal}^1$ and $D_{cal}^2$ with sizes $|D_{cal}^1| = 2,000$ and $|D_{cal}^2| = 8,000$. We conduct 1,000 Monte Carlo simulations using $D_{test}$. As we discussed in Section 6, the methods Sequential CP, E2E-CP with ARA, and E2E-CP with IRA are analyzed.

Table 2: Average cost, computation time, and collision avoidance rate using the kinematic vehicle model with different methods.

| | | Sequential CP | E2E-CP | |
| | | | with ARA | with IRA |
| --- | --- | --- | --- | --- |
| Average cost | $\alpha = 0.05$ | 22.20 | 20.46 | 4.77 |
| | $\alpha = 0.10$ | 20.24 | 18.78 | 3.52 |
| | $\alpha = 0.15$ | 19.22 | 17.35 | 3.18 |
| | $\alpha = 0.20$ | 17.05 | 15.13 | 2.89 |
| Average computation time | $\alpha = 0.05$ | 0.111 | 0.100 | 0.128 |
| | $\alpha = 0.10$ | 0.093 | 0.087 | 0.126 |
| | $\alpha = 0.15$ | 0.078 | 0.085 | 0.130 |
| | $\alpha = 0.20$ | 0.076 | 0.078 | 0.131 |
| Collision avoidance rate | $\alpha = 0.05$ | 95.4% | 95.7% | 98.4% |
| | $\alpha = 0.10$ | 93.9% | 93.1% | 98.3% |
| | $\alpha = 0.15$ | 90.8% | 89.8% | 97.6% |
| | $\alpha = 0.20$ | 88.4% | 89.1% | 91.2% |

Table 2 shows the average cost, average computation time, and collision avoidance rate of 1,000 simulations using the kinematic vehicle model with different methods. We collect the simulation data under different total risk tolerances $\alpha = 0.05, 0.10, 0.15, 0.20$. On one hand, with an increase in total risk tolerance, the average cost of all methods decreases. On the other hand, benefiting from the feedback information of posterior probabilities, the average cost of E2E-CP with ARA shows a reduction of 7.21% to 11.26% compared to Sequential CP. Furthermore, by flexibly allocating the allowable risk provided by posterior probabilities, the average cost of E2ECP with IRA exhibits a significant reduction compared with Sequential CP. Additionally, the increase in total risk tolerance provides greater flexibility in the risk allocation of E2E-CP with IRA, resulting in a significant reduction in its average cost. As mentioned in Section 6, the calculation of posterior probabilities does not incur additional computational burden. Therefore, the average computation time of E2E-CP with ARA is essentially comparable to that of Sequential CP. The collision rates of all methods do not exceed their corresponding total risk tolerances.

## C.2 SIMULATION FOR LINEAR QUADROTOR MODEL

We examine the quadrotor model Mistler et al. (2001) with the following linear dynamics.

$$
\begin{aligned}
&\ddot{x} = g\theta && \ddot{y} = -g\phi && \ddot{z} = \frac{1}{m_Q}u_1 \\
&\ddot{\phi} = \frac{l_Q}{I_{xx}}u_2 && \ddot{\theta} = \frac{l_Q}{I_{yy}}u_3 && \ddot{\psi} = \frac{l_Q}{I_{zz}}u_4
\end{aligned}
\tag{51}
$$

where $g = 9.81$ represents the gravitational acceleration, $m_Q = 0.65$ denotes the mass, and $l_Q = 0.23$ is the distance between the quadrotor and the rotor. $I_{xx} = 0.0075$, $I_{yy} = 0.0075$, and $I_{zz} = 0.013$ correspond to the area moments of inertia about the principle axes in the body frame. The states are the position and orientation with the corresponding velocities and rates — $(x, y, z, \dot{x}, \dot{y}, \dot{z}, \phi, \theta, \psi, \dot{\phi}, \dot{\theta}, \dot{\psi}) \in \mathbb{R}^{12}$. The control inputs $u_1, u_2, u_3, u_4$ correspond to the thrust force in the body frame and three moments. The system (51) is discretized using the sampling time $\Delta = 0.125$, and the total time is also set to $T = 20$.

Similar to the experiments based on the kinematic vehicle model in Appendix C.1, the objective is to control the quadrotor to reach the target point $p_{tar}$ while navigating around $M = 3$ moving obstacles. The target point constraint and obstacle avoidance constraints are consistent with those used in the simulation using the kinematic vehicle model. We randomly generate 13,000 obstacle trajectories and assign them as in Appendix C.1. The methods Sequential CP, E2E-CP with ARA, and E2E-CP with IRA are analyzed through 1,000 Monte Carlo simulations.

Table 3: Average cost, computation time, and collision avoidance rate using the quadrotor model with different methods.

| | | Sequential CP | E2E-CP | |
| | | | with ARA | with IRA |
|---|---|---|---|---|
| Average cost | $\alpha = 0.05$ | 17.321 | 15.356 | 7.189 |
| | $\alpha = 0.10$ | 16.168 | 14.228 | 6.798 |
| | $\alpha = 0.15$ | 14.835 | 12.354 | 6.191 |
| | $\alpha = 0.20$ | 13.217 | 10.222 | 5.398 |
| Average computation time | $\alpha = 0.05$ | 0.022 | 0.027 | 0.038 |
| | $\alpha = 0.10$ | 0.020 | 0.021 | 0.039 |
| | $\alpha = 0.15$ | 0.021 | 0.020 | 0.037 |
| | $\alpha = 0.20$ | 0.020 | 0.019 | 0.036 |
| Collision avoidance rate | $\alpha = 0.05$ | 98.8% | 98.2% | 96.3% |
| | $\alpha = 0.10$ | 93.5% | 94.6% | 94.1% |
| | $\alpha = 0.15$ | 92.0% | 90.2% | 91.9% |
| | $\alpha = 0.20$ | 88.2% | 86.7% | 88.2% |

Table 3 shows the average cost, average computation time, and collision avoidance rate of 1,000 simulations using the quadrotor model with different methods. The experimental results using the quadrotor model are fundamentally consistent with those derived from the experiments using the kinematic vehicle model. Compared with Sequential CP, E2E-CP with ARA benefits from the posterior probabilities calculation, leading to a moderate improvement in performance. E2E-CP with IRA, leveraging the combined use of posterior probabilities and a more flexible risk allocation, exhibits a significant enhancement in performance. Note that due to the linear nature of the quadrotor model, there is a significant reduction in computation time compared to the nonlinear vehicle model.

## C.3 SIMULATION FOR DYNAMIC BICYCLE MODEL

We examine a vehicle with the following dynamic bicycle model Hakobyan & Yang (2021).

$$\dot{x} = v_x \cos\theta - v_y \sin\theta \tag{52}$$

$$\dot{y} = v_x \sin\theta + v_y \cos\theta \tag{53}$$

$$\dot{\theta} = r \tag{54}$$

$$\dot{v}_y = \frac{-2(C_f + C_r)}{m_V v_x} v_y - \left( \frac{2l_f C_f - 2l_r C_r}{m_V v_x} + v_x \right) r + \frac{2C_f}{m_V} \delta_f \tag{55}$$

$$\dot{r} = \frac{-2(l_f C_f + l_r C_r)}{I_z v_x} v_y - \frac{2l_f^2 C_f - 2l_r^2 C_r}{I_z v_x} r + \frac{2l_f C_f}{I_z} \delta_f \tag{56}$$

where $x, y$ are the vehicle's central of mass, $\theta, v_y$, and $r$ are lateral velocity, orientation, and yaw rate, respectively. Furthermore, $v_x$ is the constant longitudinal velocity, $m_V$ denotes the mass of the vehicle, $C_f$ and $C_r$ represent the cornering stiffness coefficients of the front and rear tires respectively, $L_f$ and $L_r$ denote the distances from the center of mass to the front and rear wheels, and $I_z$ corresponds to the moment of inertia around the $z$-axis. The input variable is the front wheel steering angle $\delta_f$. The system (52) is discretized using the sampling time $\Delta = 0.125$, and the total time is also set to $T = 20$. According to Hakobyan & Yang (2021), the parameters of the dynamic bicycle model used in this simulation are listed in Table 4.

Table 4: Dynamic bicycle model parameters.

| $m_V$ | $C_f$ | $C_r$ | $I_z$ | $L_f$ | $L_r$ | $v_x$ |
|---|---|---|---|---|---|---|
| $1700kg$ | $50kN/rad$ | $50kN/rad$ | $6000kg \cdot m^2$ | $1.2m$ | $1.3m$ | $5m/s$ |

The task is to steer the vehicle to its target point $p$ while avoiding $M = 2$ moving obstacles. Similar to the experiments in Appendix C.1, the target point constraint and obstacle avoidance constraints are incorporated into the optimization problem to ensure the vehicle reaches the target point while avoiding collisions with obstacles. We collect 13,000 joint obstacle trajectories and assign them as in Appendix C.1. The methods Sequential CP, E2E-CP with ARA, and E2E-CP with IRA are analyzed through 1,000 Monte Carlo simulations.

Table 5: Average cost, computation time, and collision avoidance rate using the dynamic bicycle model with different methods.

| | | Sequential CP | E2E-CP | |
| | | | with ARA | with IRA |
| --- | --- | --- | --- | --- |
| Average cost | $\alpha = 0.05$ | 23.05 | 20.91 | 13.77 |
| | $\alpha = 0.10$ | 22.38 | 18.39 | 11.35 |
| | $\alpha = 0.15$ | 20.71 | 16.99 | 10.17 |
| | $\alpha = 0.20$ | 16.55 | 14.78 | 8.58 |
| Average computation time | $\alpha = 0.05$ | 0.365 | 0.361 | 0.884 |
| | $\alpha = 0.10$ | 0.339 | 0.335 | 0.817 |
| | $\alpha = 0.15$ | 0.494 | 0.506 | 1.292 |
| | $\alpha = 0.20$ | 0.309 | 0.407 | 1.003 |
| Collision avoidance rate | $\alpha = 0.05$ | 96.8% | 96.5% | 97.0% |
| | $\alpha = 0.10$ | 94.8% | 94.0% | 94.3% |
| | $\alpha = 0.15$ | 91.5% | 90.0% | 91.5% |
| | $\alpha = 0.20$ | 89.5% | 87.8% | 89.5% |

Table 5 shows the average cost, average computation time, and collision avoidance rate of 1,000 simulations using the dynamic bicycle model with different methods. The experimental results are generally consistent with those obtained from the experiments using the kinematic vehicle mode and the quadrotor model. The performance of E2E-CP shows a certain degree of improvement over Sequential CP based on posterior probability calculations. Based on posterior probability calculations, E2E-CP with ARA demonstrates a certain level of performance improvement compared to Sequential CP, while E2E-CP with IRA further attains significant performance by leveraging the combined use of posterior probabilities and a more flexible risk allocation. It should be noted that, due to the simulation of a relatively complex nonlinear model in this experiment, the average computation time inevitably increases. Furthermore, it may be observed that the reduction in average cost achieved by E2E-CP with IRA compared to Sequential CP decreases in this experiment (47.2% reduction) compared with the experiment using the kinematic vehicle model in Appendix C.1 (81.9% reduction). This is because, compared to relatively simple scenarios (2 obstacles, dynamic bicycle model experiment), more complex scenarios (3 obstacles, kinematic vehicle model experiment) better highlight the performance improvements enabled by the flexibility in risk allocation.

In summary, the three simulation experiments demonstrate the general applicability of the proposed method, achieving significant performance improvements across various system models while satisfying probabilistic collision avoidance requirements. In fact, the complexity of different system models only affects the average computation time. In addition, simulations demonstrate that E2E-CP with IRA achieves more significant performance improvements in relatively complex scenarios.

## D  EXPERIMENTS AND DISCUSSION ON DISTRIBUTION SHIFT

The individual chance constraint reformulation in Lemma 4.1 and the posterior probability calculation in Lemma 4.2 rely on Assumptions 3.1 and 3.2, which imply that real joint obstacle trajectory and those in the training and calibration datasets follow the same distribution $\mathcal{D}$. Specifically, Assumption 3.1 posits that the system does not influence the real joint obstacle trajectory, which holds approximately in many robotic applications, e.g., autonomous vehicles behave in ways that result in socially acceptable trajectories that do not change the behavior of pedestrians Lindemann et al. (2023). Assumption 3.2 assumes the availability of the training and calibration datasets. Assumption 3.2 is commonly used and not restrictive in practice Sun & Yu (2023); Stankeviciute et al. (2021), as extensive data can be sourced from advanced high-fidelity simulators or robotic applications like autonomous vehicles, where datasets are increasingly accessible.

Although many scenarios approximately satisfy our assumption of the same distribution, we acknowledge that the system states $x$ may change $\mathcal{D}$ during test time, e.g., when a robot is too close to a pedestrian. However, in this appendix, we demonstrate through analysis and experiments that the proposed method exhibits a certain degree of robustness to moderate distribution shifts. Specifically,

we design experiments to compare the effects of different levels of distribution shifts between test trajectories and calibration trajectories on the performance and safety of the proposed method.

Apart from the method of generating obstacle trajectories, the experimental setup is identical to that of the kinematic vehicle model experiment in Appendix C.1. To obtain obstacle trajectories with different distributions, the obstacles are modeled using the following double integrator model.

$$
\begin{bmatrix} p_{x,t+1} \\ p_{y,t+1} \\ v_{x,t+1} \\ v_{y,t+1} \end{bmatrix} = \begin{bmatrix} p_{x,t} + \Delta v_{x,t} + \frac{\Delta^2}{2} a_{x,t} \\ p_{y,t} + \Delta v_{y,t} + \frac{\Delta^2}{2} a_{y,t} \\ v_{x,t} + \Delta a_{x,t} \\ v_{y,t} + \Delta a_{y,t} \end{bmatrix} \tag{57}
$$

where $(p_x, p_y, v_x, v_y)$ is the state of an obstacle, consisting of its center of mass and velocity vector. The control input $u = (a_x, a_y)$ is the acceleration vector. Similarly, the sampling time $\Delta$ is selected as $0.125$. The obstacle trajectories from a given start point to the target point are obtained by solving an optimization problem. And the obstacle trajectories with different distributions are generated by adding zero-mean Gaussian noise $\mathcal{N}(0, \sigma^2)$ with varying covariance $\sigma$ to the system input $u$. Specifically, the trajectories in the training and calibration datasets are generated under $\sigma_{cali} = 0.3$, while the test trajectories are generated under different values of $\sigma_{test}$. Thus, the difference between $\sigma_{test}$ and $\sigma_{cali}$ reflects the magnitude of the distributional shift. For each different value of $\sigma_{test}$, we conduct 1,000 Monte Carlo experiments.

Table 6: Average cost and collision avoidance rate using the kinematic vehicle model with different distribution shifts ($\alpha = 0.2$).

| | | Sequential CP | E2E-CP | |
| --- | --- | --- | --- | --- |
| | | | with ARA | with IRA |
| Average cost | $\sigma_{test} = 0.01$ | 15.45 | 12.36 | 3.88 |
| | $\sigma_{test} = 0.10$ | 15.98 | 13.70 | 3.96 |
| | $\sigma_{test} = 0.30$ | 17.27 | 15.82 | 5.06 |
| | $\sigma_{test} = 1.00$ | 20.53 | 16.66 | 7.27 |
| | $\sigma_{test} = 2.00$ | 22.78 | 18.93 | 9.77 |
| | $\sigma_{test} = 3.00$ | 23.93 | 20.72 | 10.75 |
| Collision avoidance rate | $\sigma_{test} = 0.01$ | 97.6% | 99.0% | 99.6% |
| | $\sigma_{test} = 0.10$ | 97.6% | 97.9% | 99.6% |
| | $\sigma_{test} = 0.30$ | 90.4% | 90.0% | 92.4% |
| | $\sigma_{test} = 1.00$ | 85.7% | 85.7% | 84.1% |
| | $\sigma_{test} = 2.00$ | 83.6% | 82.8% | 82.4% |
| | $\sigma_{test} = 3.00$ | 79.3% | 78.9% | 78.2% |

Table 6 shows the average cost and collision avoidance rate of 1,000 simulations using the kinematic vehicle model with different values of $\sigma_{test}$. An increase in $\sigma_{test}$ relative to $\sigma_{cali}$ indicates an increase in the distributional shift between the test and calibration trajectories. However, even when $\sigma_{test}$ increases to 2, the collision rate of the proposed method remains within the total risk tolerance ($\alpha = 0.2$). Although the collision rate no longer meets the risk tolerance requirement when $\sigma_{test}$ increases to 3, we will demonstrate in Appendix E that such large distribution shifts are unlikely to occur in practical applications. Moreover, an increase in $\sigma_{test}$ leads to unexpected obstacle movements, which ultimately result in an increase in the average cost. However, under all values of $\sigma_{test}$, the proposed method achieves a significant decrease in average cost compared to the Sequential CP. In summary, the proposed method exhibits a certain degree of robustness to moderate distribution shifts. Specifically, when the distribution shift between the test trajectory and the calibration trajectories is not substantial, the proposed method can maintain the satisfaction of the total risk constraint and the performance improvement compared to Sequential CP.

One might question whether, in realistic scenarios, the test trajectory could experience a large distribution shift, leading to a violation of the total risk constraints. Fortunately, this is highly unlikely to occur in practice. In Strawn et al. (2023), the author explicitly checked through experiments that interactions among agents in multi-agent systems do not introduce large distribution shifts.

# E   EXPERIMENTS AND DISCUSSION ON DEPENDENT SCENARIOS

In this appendix, we directly design simulation scenarios where the system and obstacles are dependent, to demonstrate the safety and high performance of the proposed method in realistic scenarios.

Except for the method of generating obstacle trajectories, the experimental setup is identical to that of the kinematic vehicle model experiment in Appendix C. The obstacles are modeled using the double integrator model (57). The training and calibration trajectories are generated in the same manner as described in Appendix D. Considering the interdependence between the system and the obstacles, the test obstacle trajectories are generated online during test time based on the current system state. Specifically, only the references of the test trajectories are generated offline. The real test trajectories are obtained by solving an optimization problem to follow the reference trajectories. Additionally, when obstacles approach the system, the distance between the obstacles and the system is incorporated into the objective function to simulate real-world avoidance behavior. Both the scenarios, considering and ignoring the interdependence between the system and the obstacles, are simulated.

Table 7: Average cost and collision avoidance rate using the kinematic vehicle model with independent and dependent obstacles.

| | | Sequential CP | E2E-CP | |
| | | | with ARA | with IRA |
|---|---|---|---|---|
| | | $\alpha = 0.1$ | | |
| Average cost | Independent | 20.116 | 18.022 | 4.053 |
| | Dependent | 20.269 | 18.005 | 4.123 |
| Collision avoidance rate | Independent | 94.7% | 93.6% | 94.4% |
| | Dependent | 93.8% | 91.0% | 93.3% |
| | | $\alpha = 0.2$ | | |
| Average cost | Independent | 17.289 | 14.932 | 3.032 |
| | Dependent | 17.279 | 15.007 | 3.045 |
| Collision avoidance rate | Independent | 88.2% | 89.5% | 92.0% |
| | Dependent | 86.5% | 88.3% | 89.5% |

Table 7 shows the average cost and collision avoidance rate of 1,000 simulations using the kinematic vehicle model with independent and dependent obstacles. "Independent" and "Dependent" represent the cases where the influence of the system on the obstacle trajectories is disregarded and considered, respectively. It can be observed that the influence of the system on obstacles indeed reduces the overall collision avoidance rate, as it disrupts the exchangeability between the test and calibration trajectories. However, the reduction in the collision avoidance rate is negligible and remains well within the corresponding total risk tolerance. Moreover, Table 7 also shows that the dependence between the system and obstacles has almost no impact on the average cost. In summary, in realistic scenarios, relaxing Assumption 3.1 does not significantly impact the performance of the proposed algorithm.

# F   DETAILS ABOUT THE PREDICTION REGIONS

Table 8 shows the prediction region radius for different time $t$ and $\tau$ ($C_{\tau|t}$) using the kinematic vehicle model with different methods (Sequential CP and E2E-CP with ARA). At the initial state ($t = 0$), no realized state is available for calculating posterior probabilities. As a result, $C_{\tau|0}$ for all $\tau$ obtained by Sequential CP and E2E-CP with ARA are essentially identical, with minor differences arising from the fact that Sequential CP utilizes the calibration $\mathcal{D}_{cal}$, whereas E2E-CP with ARA only employs $\mathcal{D}_{cal}^1$. As the system operates, an increasing number of realized states $x_t^*$ are available for the calculation of posterior probabilities, enabling E2E-CP to yield a relatively narrower prediction region, corresponding to a smaller $C_{\tau|t}$. As shown in Table 8, the average ratio of the predicted region radius obtained by E2E-CP with ARA to those by Sequential CP generally exhibits a decreasing trend as time $t$ increases. Moreover, when $t > 10$, based on sufficient posterior

probabilities, the prediction region radius obtained by E2E-CP with ARA is reduced by more than $50\%$ compared to that of Sequential CP.

Table 8: Prediction region radius for different $\tau$ and $t$ ($C_{\tau|t}$) using the kinematic vehicle model with different methods (Sequential CP and E2E-CP with ARA) across 1,000 simulations ($\alpha = 0.2$).

| | | $\tau = 3$ | $\tau = 6$ | $\tau = 9$ | $\tau = 12$ | $\tau = 15$ | $\tau = 18$ | Ratio |
|---|---|---|---|---|---|---|---|---|
| $t = 0$ | E2E-CP with ARA | 0.125 | 0.258 | 0.374 | 0.519 | 0.698 | 0.913 | 1.005 |
| | Sequential CP | 0.129 | 0.247 | 0.374 | 0.520 | 0.691 | 0.902 | |
| $t = 3$ | E2E-CP with ARA | ╲ | 0.102 | 0.216 | 0.351 | 0.527 | 0.705 | 0.793 |
| | Sequential CP | | 0.129 | 0.263 | 0.409 | 0.598 | 0.738 | |
| $t = 6$ | E2E-CP with ARA | ╲ | ╲ | 0.141 | 0.289 | 0.442 | 0.604 | 0.956 |
| | Sequential CP | | | 0.152 | 0.304 | 0.451 | 0.625 | |
| $t = 9$ | E2E-CP with ARA | ╲ | ╲ | ╲ | 0.134 | 0.268 | 0.414 | 0.889 |
| | Sequential CP | | | | 0.149 | 0.305 | 0.466 | |
| $t = 12$ | E2E-CP with ARA | ╲ | ╲ | ╲ | ╲ | 0.059 | 0.139 | 0.453 |
| | Sequential CP | | | | | 0.138 | 0.291 | |
| $t = 15$ | E2E-CP with ARA | ╲ | ╲ | ╲ | ╲ | ╲ | 0.054 | 0.470 |
| | Sequential CP | | | | | | 0.115 | |

## G    LIMITATIONS

The proposed E2E-CP has two limitations. The first limitation lies in the reliance of the proposed method on the size of the calibration dataset. As previously mentioned, to ensure coverage guarantees within the end-to-end framework, the calibration dataset needs to be split into two parts: one for forward computation of prediction regions and the other for backward computation of posterior probabilities. This requirement results in the proposed method needing a larger calibration dataset compared to standard CP methods. However, extensive data can be sourced from advanced high-fidelity simulators or robotic applications like autonomous vehicles, where datasets are increasingly accessible. Thus we believe that the reliance on data quantity will not present a substantial challenge.

Additionally, the second limitation lies in the reliance of the proposed method's theoretical guarantees on exchangeability. Although we have empirically demonstrated that the proposed method exhibits a certain degree of robustness to moderate distribution shifts (Appendix D) and ensures safety and high performance in the realistic setting (Appendix E), we do not address distribution shifts in full generality or provide theoretical guarantees. Therefore, extending the proposed method beyond exchangeability represents a promising direction for future work. To this end, we propose several potential ways to extend the proposed method beyond the assumption of exchangeability. Firstly, the robust conformal prediction Cauchois et al. (2024) can be used to obtain valid prediction regions for all distributions that are "close" to $\mathcal{D}$ (in terms of the f-divergence), and integrate these in our algorithm. Additionally, integrating the adaptive conformal prediction Gibbs & Candes (2021) with our proposed method is also a potentially viable approach. Moreover, updating the calibration dataset online using real-time data based on a sliding window Xu & Xie (2021) may also be a potential way to go beyond the assumption of exchangeability.

