# OpenReview forum: "End-to-End Conformal Prediction for Trajectory Optimization"
_ICLR.cc/2025/Conference — Submitted to ICLR 2025_

### Official Review · Reviewer_iGBg · 2024-11-03

**Soundness:** 4
**Presentation:** 4
**Contribution:** 4
**Rating:** 8
**Confidence:** 3

**Summary:**

This paper studies the problem of online trajectory optimization in an uncertain environment. The proposed algorithm includes a chance-constrained optimization problem that applies conformal prediction (CP) in two ways. First, it uses forward-looking CP to predict a priori confidence intervals for future obstacle positions, which are used for planning. Second, it uses backward-looking CP to compute the posterior confidence intervals of past obstacle positions to estimate how much of the risk budget was actually "used up" compared to the planned amount of risk. The risk budget freed up by this backward-looking assessment is applied to future timesteps, allowing less conservative behavior. The experiments are simple, presenting well-chosen ablations rather than a comprehensive suite of baseline methods, but I found them to be compelling.

**Strengths:**

I really enjoyed reading this paper. The core of the paper is a really interesting idea (understand how much risk you have previously incurred, so as to better decide how much risk to take on in future), and the authors rigorously build a rigorous framework around this idea to design an intuitive trajectory optimization framework.

The paper is clearly written throughout, and although I did not check all proofs, the ones I did check were fairly clear. It is possible that there is some nuance of conformal prediction that I am missing, but overall the approach seems sound.

**Weaknesses:**

The paper is missing some key implementation details (notably: how was the optimization problem solved/what solver was used?). The paper is also missing a discussion of limitations. If there is not space in the main paper, these could be included in the appendix, but I would like to see limitations at least mentioned in the main text.

**Questions:**

1. In section 3.2, is the prediction also conditioned on the history of obstacle positions $Y_{0:t}$ or only on the most recent observation $Y_t$? Your notation implies the latter, but I'd imagine that the former would yield more accurate predictions.
2. Line 90: "level $\beta$ quantile" -- do you mean "level $1-\alpha$ quantile"? If not, what is $\beta$?
3. The empirical results suggest that the proposed method (E2E-CP with IRA) is still conservative. What do you think are the reasons for this, and how would you go about reducing this conservatism?

---

> ### Author Response · Authors · 2024-11-21
> **Rebuttal by Authors – Part I**
>
> We thank the reviewer for the constructive comments! We have revised the manuscript to address the concerns.
>
> >**Weakness:** The paper is missing some key implementation details (notably: how was the optimization problem solved/what solver was used?). The paper is also missing a discussion of limitations. If there is not space in the main paper, these could be included in the appendix, but I would like to see limitations at least mentioned in the main text.
>
> Thank you very much for your valuable comment. Following the reviewer’s valuable suggestion, we have added implementation details in Section 6 of the revised manuscript, indicating that the solver IPOPT (v3.12.9) was used to solve the optimization problem (line 485).
>
> Additionally, in the revised manuscript, we have also added the discussion of limitations in Section 7 of the main text. Specifically, the proposed method has two limitations. The first limitation lies in the reliance of the proposed method on the size of the calibration dataset. The proposed method requires splitting the calibration dataset into two parts, which results in the need for a larger calibration dataset compared to standard CP methods. The second limitation lies in the reliance of the proposed method’s theoretical guarantees on exchangeability. Although we have empirically demonstrated that the proposed method exhibits a certain degree of robustness to moderate distribution shifts (Appendix D), we do not address distribution shifts in full generality or provide theoretical guarantees.
>
> In addition to the discussion on limitations, we have also proposed several potential ways to address the limitations in Appendix G of the revised manuscript.

---

> ### Author Response · Authors · 2024-11-21
> **Rebuttal by Authors – Part II**
>
> >**Question 1:** In section 3.2, is the prediction also conditioned on the history of obstacle positions $Y_{0:t}$ or only on the most recent observation $Y_{t}$? Your notation implies the latter, but I'd imagine that the former would yield more accurate predictions.
>
> Thank you very much for your valuable comment. In our implementation, the prediction is conducted using the history of obstacle positions $Y_{0:t}$. We have corrected the typo in the revised manuscript (lines 162-174).

---

> ### Author Response · Authors · 2024-11-21
> **Rebuttal by Authors – Part III**
>
> >**Question 2:** Line 90: "level $\beta$ quantile" -- do you mean "level $1-\alpha$ quantile"? If not, what is $\beta$?
>
> Thank you very much for your valuable comment. This should indeed be $1-\alpha$. We have corrected this minor typo in the revised manuscript (line 190).

---

> ### Author Response · Authors · 2024-11-21
> **Rebuttal by Authors – Part IV**
>
> >**Question 3:** The empirical results suggest that the proposed method (E2E-CP with IRA) is still conservative. What do you think are the reasons for this, and how would you go about reducing this conservatism?
>
> Thank you very much for your valuable comment. We believe that one source of conservatism lies in the reformulation of the joint chance constraint using the Boole's inequality (5) (lines 214-215). The Boole's inequality utilizes the upper bound of the union, representing a rather coarse approximation that does not account for the correlation of obstacle positions in two time steps. Conservatism can be reduced by replacing the Boole's inequality with the computation of the joint risk posterior given the realized obstacle positions or with adaptive methods.

---

> > ### Comment · Reviewer_iGBg · 2024-11-24
> >
> > Thank you for your response. I have read the response to my review and am satisfied with the authors answers. I believe my original rating (8 - accept, good paper) is still appropriate.

---

> > > ### Author Response · Authors · 2024-11-24
> > > **Thank you very much for your kind reply.**
> > >
> > > Thank you so much for your kind reply and your value comments, which have helped us to improve the quality of this manuscript. Again, we sincerely thank you for supporting our work.

---

### Official Review · Reviewer_t4JB · 2024-11-03

**Soundness:** 3
**Presentation:** 3
**Contribution:** 2
**Rating:** 6
**Confidence:** 5

**Summary:**

In this paper, the authors proposed a conformal prediction based posterior risk calculation method to obtain the posterior probability of collision conditional on realized trajectories. This method leverages the realized trajectories to adjust the posterior allowable risk in an online setting. The experiment results demonstrate the efficacy of the proposed method.

**Strengths:**

1) The paper is well organized and easy to follow.
2) This paper utilizes the conformal prediction-based method to achieve the end-to-end adjustments of the prediction region, improving the limitation of conformal prediction.

**Weaknesses:**

1) The prediction regions should be narrow and cover the ground truth data. However, large prediction regions would also achieve a high collision avoidance rate but not meaningful in real-world situations. The authors should provide more details about the prediction region.
2) Proving the effectiveness of the proposed method is very important. However, the authors merely apply the proposed method on two trajectory prediction models which were proposed in 2001 and 2006, respectively. The authors should provide experiment results on more state-of-the-art models.

**Questions:**

1) Performance of the proposed method on more state-of-the-art methods.
2) Performance of the proposed method under data distribution shift.

---

> ### Author Response · Authors · 2024-11-21
> **Rebuttal by Authors – Part I**
>
> We thank the reviewer for the constructive comments! We have revised the manuscript to address the concerns.
>
> >**Weakness 1:** The prediction regions should be narrow and cover the ground truth data. However, large prediction regions would also achieve a high collision avoidance rate but not meaningful in real-world situations. The authors should provide more details about the prediction region.
>
> Thank you very much for your valuable comment. The proposed method actually provides theoretical guarantees for both the coverage and efficiency (narrowness) of the prediction region. On one hand, the coverage guarantee of the prediction region is provided in Lemma 3.1 of the original manuscript. On the other hand, Corollary 4.1 restricts the upper bound of the expectation of posterior probability. This ensures that the prediction region obtained by the proposed method is at least narrower than the one obtained through the state-of-the-art sequential method [1], and its efficiency can be continuously improved during trajectory optimization. In fact, the ability to achieve the narrowing of the prediction region while ensuring coverage is precisely the advantage and contribution of the proposed method.
>
> Following the reviewer’s comments, we provide more details about how narrow the prediction regions are through experiments. Table 8 (Appendix F) presents the prediction region radius for different time $t$ and $\tau$ ($C_{\tau|t}$) with different methods. Since iterative risk allocation affects the prediction region radius, we only compared $C_{\tau|t}$ obtained through E2E-CP with ARA and Sequential CP to make a fair comparison. At the initial state ($t=0$), no realized state is available for calculating posterior probabilities. As a result, $C_{\tau|0}$ for all $\tau$ obtained by Sequential CP and E2E-CP with ARA is essentially identical. As the system operates, an increasing number of realized states $x_{t}^{*}$ are available for the calculation of posterior probabilities, enabling E2E-CP to yield a relatively narrower prediction region, i.e. a smaller $C_{\tau|t}$. Moreover, when $t>10$, based on sufficient posterior probabilities, the prediction region radius obtained by E2E-CP with ARA is reduced by more than $50$% compared to that of Sequential CP. The reviewer can find a detailed discussion on the narrowness of the prediction region in Appendix F and lines 500 to 512 of the revised manuscript.
>
> [1] *Lars Lindemann, Matthew Cleaveland, Gihyun Shim, and George J Pappas. Safe planning in dynamic environments using conformal prediction. IEEE Robotics and Automation Letters. 2023.*

---

> ### Author Response · Authors · 2024-11-21
> **Rebuttal by Authors – Part II**
>
> >**Weakness 2:** Proving the effectiveness of the proposed method is very important. However, the authors merely apply the proposed method on two trajectory prediction models which were proposed in 2001 and 2006, respectively. The authors should provide experiment results on more state-of-the-art models.
> >
> >**Question 1:** Performance of the proposed method on more state-of-the-art methods.
>
> Thank you very much for your valuable comment. First, we would like to clarify that the primary contribution of this paper is to propose the E2E-CP - a novel end-to-end Conformal Prediction (CP) method for Trajectory Optimization (TO), rather than a novel system model. Therefore, in the experiments of the original manuscript, we compared our method with the state-of-the-art CP-based TO method (2023) [1] to demonstrate the performance of our method.
>
> Additionally, the two robotic models employed in our experiments of the original manuscript—the kinematic vehicle model and the quadrotor model—are classical robotic models and serve as mainstream benchmarks for evaluating the performance of algorithms in trajectory optimization, trajectory planning, and motion control [1][2][3]. Many systems in daily life (car-like wheeled mobile agents, unicycles, etc) can be modeled as the kinematic vehicle model which is used in our paper.
>
> Finally, we fully agree with the reviewer that testing our algorithm on additional models would provide a more comprehensive evaluation of its performance. Following the reviewer’s valuable suggestion, we implemented our algorithm on the dynamic bicycle model (2021) [2], and the experimental results and details are provided in Appendix C.3 of the revised manuscript. The experimental results show that, under this model, the proposed method achieves at least a $40.6$% reduction in average cost compared to the state-of-the-art method, further demonstrating the superiority and generalizability of the proposed method. The reviewer can find the detailed results (Table 5) and the analysis of the newly added experiment in Appendix C.3 of the revised manuscript.
>
> [1] *Lars Lindemann, Matthew Cleaveland, Gihyun Shim, and George J Pappas. Safe planning in dynamic environments using conformal prediction. IEEE Robotics and Automation Letters. 2023.*
>
> [2] *Astghik Hakobyan and Insoon Yang. Wasserstein distributionally robust motion control for collision avoidance using conditional value-at-risk. IEEE Transactions on Robotics. 2021.*
>
> [3] *Bolun Dai, Rooholla Khorrambakht, Prashanth Krishnamurthy, Vinicius Goncalves, Anthony Tzes, and Farshad Khorrami. Safe navigation and obstacle avoidance using differentiable optimization based control barrier functions. IEEE Robotics and Automation Letters. 2023.*

---

> ### Author Response · Authors · 2024-11-21
> **Rebuttal by Authors – Part III**
>
> >**Question 2:** Performance of the proposed method under data distribution shift.
>
> Thank you very much for your valuable comment. Following the reviewer’s valuable suggestion, we have added experiments evaluating the proposed method under varying degrees of distribution shift (Appendix D) and scenarios where the system and obstacles are dependent (Appendix E).
>
> For the experiments under distribution shift, we generate test trajectories with varying degrees of distribution shifts by applying zero-mean Gaussian noise with varying covariance to the accelerations of the obstacles. The experimental results indicate that under moderate distribution shifts, the proposed method can still satisfy the total risk tolerance constraint while maintaining an average cost reduction of at least $57.1$%. The reviewer can find the detailed results (Table 6) and the analysis of the experiment in Appendix D of the revised manuscript.
>
> We would like to further clarify that when the distribution shifts become excessively large, the proposed method inevitably exceeds the total risk tolerance. However, reference [1] explicitly checked through experiments that interactions among agents in multi-agent systems do not introduce large distribution shifts. Furthermore, we directly design simulation scenarios where the obstacles’ avoidance behavior for the system (beyond Assumption 3.1) is incorporated. The experimental results indicate that the reduction in the collision avoidance rate due to the dependence between the system and obstacles is negligible. The reviewer can find the detailed results (Table 7) and the analysis of the experiment in Appendix E of the revised manuscript.
>
> In summary, the robustness of our method to distribution shifts is sufficient to handle realistic scenarios. Additionally, in Appendix G of the revised manuscript, we discuss several potential ways to address the distribution shift in full generality or to provide theoretical guarantees.
>
> [1] *Kegan J Strawn, Nora Ayanian, and Lars Lindemann. Conformal predictive safety filter for rl controllers in dynamic environments. IEEE Robotics and Automation Letters. 2023.*

---

> ### Author Response · Authors · 2024-12-01
> **Looking forward to your feedback. Many thanks!**
>
> Dear Reviewer t4JB,
>
> Again, thank you so much for your previous valuable comments. We would like to follow up on our rebuttal to ensure that all your concerns have been adequately addressed. If there are any further questions, we will be very glad to address them. Your feedback is extremely valuable in helping us improve our manuscript, and we eagerly await your reply.
>
> Thank you very much for your time and consideration.
>
> Best regards,
>
> The Authors

---

### Official Review · Reviewer_wqrY · 2024-11-04

**Soundness:** 3
**Presentation:** 3
**Contribution:** 2
**Rating:** 5
**Confidence:** 4

**Summary:**

This paper introduces a novel trajectory optimization algorithm grounded in conformal prediction, which dynamically adjusts posterior risk based on observed trajectories. Benchmark experiments highlight the algorithm’s effectiveness, showcasing its superior performance compared to existing methods.

**Strengths:**

1. The algorithm proposed in the paper is novel, supported by rigorous theorems and proofs that strengthen its theoretical foundation.
2. The presentation is clear and well-structured.

**Weaknesses:**

1. I have concerns regarding the assumptions made in this paper, particularly the strong independence assumptions on obstacle trajectories, which form the basis of the proposed algorithm. In Section 3.1, the authors assume the independence between D and system (1) (lines 134-135) and further assume that the real joint obstacle trajectory Y and the N available joint obstacle trajectories Y^(i) are independent and identically distributed (i.i.d.) (lines 143-145). However, in real-world applications such as autonomous driving, obstacle trajectories—specifically those of other vehicles on the road—are highly interdependent, as each vehicle’s decisions are influenced by the behaviors of surrounding vehicles. The interdependence between obstacle trajectories is a key factor contributing to the complexity of real-world problems. Assuming independence significantly simplifies the problem but may limit the algorithm's applicability in practical, interdependent scenarios. It would be beneficial for the authors to provide a justification for these independence assumptions and to identify any real-world applications where obstacle trajectories can be reasonably assumed to be independent.
2. Another concern is the limited performance improvement observed with the proposed algorithm. In Table 1, the collision avoidance rate shows only a marginal increase, raising questions about the practical impact of the algorithm. Furthermore, in Table 3 (Appendix C.2), the collision avoidance rate actually decreases when using the proposed algorithm, suggesting potential issues with its robustness. Additionally, the benchmark experiments include only two robotic systems, which may not provide sufficient diversity to fully evaluate the algorithm's effectiveness across different scenarios.
3. Additionally, this paper’s focus on trajectory optimization is largely unrelated to machine learning or deep learning. Given the emphasis on trajectory optimization techniques rather than learning-based methods, it would be more appropriately submitted to robotics-focused publication venues rather than conferences dedicated to deep learning.
4. Minor typo: On line 190, "level beta" should be corrected to "level alpha."

**Questions:**

1. As noted in the weaknesses section, it would be valuable for the authors to provide a rationale for these independence assumptions and to highlight any real-world applications in which obstacle trajectories can reasonably be assumed to be independent.
2. The paper would benefit from additional benchmark examples beyond the two robotic systems currently evaluated. Expanding the benchmarks would provide a more comprehensive assessment of the algorithm's effectiveness across diverse scenarios.
3. It would be helpful if the authors could clarify why a paper focused on trajectory optimization, rather than on learning-based approaches, is suitable for a deep learning conference.

---

> ### Author Response · Authors · 2024-11-21
> **Rebuttal by Authors – Part I**
>
> We thank the reviewer for the constructive comments! We have revised the manuscript to address the concerns.
>
> >**Weakness 1:** I have concerns regarding the assumptions made in this paper, particularly the strong independence assumptions on obstacle trajectories, which form the basis of the proposed algorithm. In Section 3.1,......
> >
> >$\vdots$
> >
> >Assuming independence significantly simplifies the problem but may limit the algorithm's applicability in practical, interdependent scenarios. It would be beneficial for the authors to provide a justification for these independence assumptions and to identify any real-world applications where obstacle trajectories can be reasonably assumed to be independent.
> >
> >**Question 1:** As noted in the weaknesses section, it would be valuable for the authors to provide a rationale for these independence assumptions and to highlight any real-world applications in which obstacle trajectories can reasonably be assumed to be independent.
>
> Thank you very much for your valuable comment. We will first discuss the assumptions made in this paper and then clarify that these assumptions are primarily intended for deriving theoretical results and do not affect the implementation of the proposed algorithm. Furthermore, we experimentally demonstrate that in realistic scenarios, relaxing these assumptions does not significantly impact the performance of the proposed algorithm.
>
> We will now provide a detailed discussion on the validity of the two assumptions made in this paper.
> 1. Assumption 3.1 posits that the system does not influence the distribution $\mathcal{D}$ of the real joint obstacle trajectory, which holds approximately in many robotic applications, e.g., autonomous vehicles behave in ways that result in socially acceptable trajectories without changing the behavior of pedestrians [1].
> 2. Assumption 3.2 assumes the availability of a dataset $D_{cal}$ with $N$ samples. We think the reviewer may have a slight misunderstanding regarding this assumption. We would like to first clarify the composition of the dataset $D_{cal}$. The system operates in an environment with $M$ obstacles. Throughout the paper, **the $M$ obstacles are treated as a single “entity”.** Note that $Y_{t}:=(Y_{t,1},...,Y_{t,M})$ represents the **joint** obstacle position, and $Y:=(Y_{0},...,Y_{T})$ represents the **joint** obstacle trajectory. Therefore, $D_{cal}$ contains the $N$ joint obstacle trajectories $Y^{(1)},...,Y^{(N)}$. Assumption 3.2 only assumes that the $N$ joint obstacle trajectories in $D_{cal}$ are i.i.d., without requiring the $M$ obstacle trajectories within the same joint obstacle trajectory to be i.i.d.. Therefore, in practice, Assumption 3.2 is commonly used [2][3] and is not restrictive, as extensive data can be sourced from advanced high-fidelity simulators or robotic applications like autonomous vehicles, where datasets are increasingly accessible.
>
> As a result, in a given scenario, we only assume that the system does not affect the obstacles trajectories, without imposing any assumptions on the interactions between the obstacles.
>
> We emphasize that the assumptions are made to provide theoretical guarantees and do not affect the implementation of the proposed method. Furthermore, we demonstrate through experiments that in realistic scenarios, relaxing the assumptions does not significantly impact the performance of the proposed algorithm. We elaborate in detail as follows.
>
> In Appendix E of the revised manuscript, we present experiments where the system and obstacles are interdependent. Specifically, the obstacles’ avoidance behavior for the system (beyond Assumption 3.1) is incorporated to simulate realistic scenarios. The experiment results indicate that the influence of the system on obstacles indeed reduces the overall collision avoidance rate, as it disrupts the exchangeability between the test and calibration trajectories. However, the reduction in the collision avoidance rate is negligible and remains well within the corresponding total risk tolerance. Moreover, the dependence between the system and obstacles has almost no impact on the average cost empirically. In summary, the experiment results show that interactions among agents in multi-agent systems do not introduce large distribution shifts and will not significantly affect our algorithm and its performance. The reviewer can find the detailed results (Table 7) and the analysis of the experiment in Appendix E of the revised manuscript.
>
> [1] *Lars Lindemann, Matthew Cleaveland, Gihyun Shim, and George J Pappas. Safe planning in dynamic environments using conformal prediction. IEEE Robotics and Automation Letters. 2023.*
>
> [2] *Sophia Huiwen Sun, and Rose Yu. Copula Conformal prediction for multi-step time series prediction. ICLR. 2024.*
>
> [3] *Kamil˙e Stankeviciute, Ahmed Alaa, and Mihaela van der Schaar. Conformal time-series forecasting. NeurIPS. 2021.*

---

> ### Author Response · Authors · 2024-11-21
> **Rebuttal by Authors – Part II**
>
> >**Weakness 2:** Another concern is the limited performance improvement observed with the proposed algorithm. In Table 1, the collision avoidance rate shows only a marginal increase, raising questions about the practical impact of the algorithm. Furthermore, in Table 3 (Appendix C.2), the collision avoidance rate actually decreases when using the proposed algorithm, suggesting potential issues with its robustness. Additionally, the benchmark experiments include only two robotic systems, which may not provide sufficient diversity to fully evaluate the algorithm's effectiveness across different scenarios.
> >
> >**Question 2:** The paper would benefit from additional benchmark examples beyond the two robotic systems currently evaluated. Expanding the benchmarks would provide a more comprehensive assessment of the algorithm's effectiveness across diverse scenarios.
>
> Thank you very much for your valuable comment. We would like to first clarify to the reviewer that the objective of the chance-constrained optimization is to **maximize performance** subject to the probabilistic constraint, rather than to maximize the constraint satisfaction probability (i.e., the collision avoidance rate). For the chance constraint, we follow the well-recognized principle of "meeting the requirement is sufficient." In fact, solely focusing on maximizing the probability of satisfying the constraint can lead to overly conservative decisions. For example, in Table 3 (Appendix C.2), for $\alpha=0.2$, the optimal scenario is to effectively “utilize” the risk tolerance ($\alpha$), where the collision avoidance probability is maintained above $80$% while minimizing the cost.
>
> Additionally, we fully agree with the reviewer that testing our algorithm on additional models and different scenarios would provide a more comprehensive evaluation of its performance. Following the reviewer’s valuable suggestion, we have implemented our algorithm on a new model and several different scenarios and presented the results in the revised manuscript.
>
> **New model experiment (Appendix C.3):**
>
> In the revised manuscript, we validate the proposed method on the dynamic bicycle model (2021) [1]. The experimental results show that, under this model, the proposed method achieves at least a $40.6$% reduction in average cost compared to the state-of-the-art method. The proposed method demonstrates superior performance across all three robotic models (Appendix C), further demonstrating the generalizability of the proposed method. The reviewer can find the detailed results (Table 5) and the analysis of the newly added experiment in Appendix C.3 of the revised manuscript.
>
> **Different scenarios (Appendix D and Appendix E):**
>
> Note that the various interactions between the robot and obstacles can be attribute to the distribution shifts in the joint obstacle trajectories. As a result, we directly investigated the impact of different degrees of distribution shifts on the safety and performance of the proposed method through experiments in Appendix D. Specifically, we generate test trajectories with varying degrees of distribution shifts by applying zero-mean Gaussian noise with varying covariances to the accelerations of the obstacles. The experimental results indicate that under moderate distribution shifts, the proposed method can still satisfy the total risk tolerance constraint while enjoying an average cost reduction of at least $57.1$%. The reviewer can find the detailed results (Table 6) and the analysis of the experiment in Appendix D of the revised manuscript.
>
> Additionally, we have included experiments simulating realistic interactions in Appendix E, as previously mentioned in “Rebuttal by Authors - Part I”.
>
> [1] *Kegan J Strawn, Nora Ayanian, and Lars Lindemann. Conformal predictive safety filter for rl controllers in dynamic environments. IEEE Robotics and Automation Letters. 2023.*

---

> ### Author Response · Authors · 2024-11-21
> **Rebuttal by Authors – Part III**
>
> >**Weakness 3:** Additionally, this paper’s focus on trajectory optimization is largely unrelated to machine learning or deep learning. Given the emphasis on trajectory optimization techniques rather than learning-based methods, it would be more appropriately submitted to robotics-focused publication venues rather than conferences dedicated to deep learning.
> >
> >**Question 3:** It would be helpful if the authors could clarify why a paper focused on trajectory optimization, rather than on learning-based approaches, is suitable for a deep learning conference.
>
> Thank you very much for your valuable comment. We provide a detailed explanation of why this paper is well-suited for this conference as follows:
>
> 1. ICLR has a dedicated robotics section. Additionally, the conferences dedicated to machine learning have published numerous papers that apply machine learning techniques to address robotic-related problems [1][2]. Note that Conformal prediction (CP) is a powerful machine learning framework for uncertainty quantification. Our paper extends this machine learning framework to an end-to-end paradigm and applies the proposed new framework to Trajectory Optimization (TO).
> 2. The main contribution of this paper lies in the proposal of E2E-CP - a novel end-to-end CP method. The proposed E2E-CP is the first CP method that focuses on downstream decision-making. Although E2E-CP is applied to TO in this paper, it is not limited to TO but can be applied to any other safety-critical decision-making under uncertainty. Additionally, serving as a quantifier of uncertainty in machine learning, CP has gained popularity in conferences dedicated to machine learning [3][4][5][6], especially ICLR. The proposed E2E-CP establishes an information channel between the upstream CP end and the downstream decision-making end, representing a novel extension of CP. Therefore, it is suitable for publication in conferences dedicated to machine learning.
>
> [1] *Dingyuan Shi, Yongxin Tong, Zimu Zhou, Ke Xu, Zheng Wang, and Jieping Ye. Graph-constrained diffusion for end-to-end path planning. ICLR 2024.*
>
> [2] *Jiankai Sun, Yiqi Jiang, Jianing Qiu, Parth Nobel, Mykel J Kochenderfer, and Mac Schwager. Conformal prediction for uncertainty-aware planning with diffusion dynamics model. NeurIPS. 2023.*
>
> [3] *Anastasios Nikolas Angelopoulos, Stephen Bates, Adam Fisch, and Lihua Lei. Conformal risk control. ICLR. 2024.*
>
> [4] *Ant’nio Farinhas, Chrysoula Zerva, Dennis Thomas Ulmer and Andre Martins. Non-exchangeable conformal risk control. ICLR. 2024.*
>
> [5] *Sophia Huiwen Sun, and Rose Yu. Copula Conformal prediction for multi-step time series prediction. ICLR. 2024.*
>
> [6] *Soroush H. Zargarbashi, and Aleksandar Bojchevski. Conformal inductive graph neural networks. ICLR. 2024.*

---

> ### Author Response · Authors · 2024-11-21
> **Rebuttal by Authors – Part IV**
>
> >**Weakness 4:** Minor typo: On line 190, "level beta" should be corrected to "level alpha."
>
> Thank you very much for your valuable comment. We have corrected this minor typo in the revised manuscript (line 190).

---

> ### Author Response · Authors · 2024-12-01
> **Looking forward to your feedback. Many thanks!**
>
> Dear Reviewer wqrY,
>
> Again, thank you so much for your previous valuable comments. We would like to follow up on our rebuttal to ensure that all your concerns have been adequately addressed. If there are any further questions, we will be very glad to address them. Your feedback is extremely valuable in helping us improve our manuscript, and we eagerly await your reply.
>
> Thank you very much for your time and consideration.
>
> Best regards,
>
> The Authors

---

> ### Comment · Reviewer_wqrY · 2024-12-02
>
> Thank you for your detailed response. After reviewing your explanation, I have decided to maintain my score. I appreciate your effort in addressing my concerns.

---

> > ### Author Response · Authors · 2024-12-03
> > **Thank you so much for your reply**
> >
> > We sincerely thank you for the kind reply and providing valuable review comments which have helped us to improve the quality of this manuscript.

---

### Official Review · Reviewer_xXLt · 2024-11-04

**Soundness:** 3
**Presentation:** 3
**Contribution:** 2
**Rating:** 5
**Confidence:** 4

**Summary:**

In this paper, the authors proposed an E2E-CP framework for shrinking-horizon TO in uncertain environments and the collision risk over the total mission time is always constrained. The proposed method leverages the information of past decisions to adjust prediction regions in an end-to-end fashion. The experiment results demonstrate the effectiveness of the proposed model for trajectory prediction task.

**Strengths:**

1.The paper provides detailed theoretical proof for the proposed method.
2. This paper compares the model performance of sequential CP, average risk allocation setting, and iterative risk allocation setting.

**Weaknesses:**

1.The prediction regions are expected to cover the ground truth trajectory and narrow. Otherwise, overlarge prediction regions would also achieve a high collision avoidance rate. The authors should provide more details about how narrow the prediction regions are.

2. The authors should demonstrate the model performance by implementing more state-of-the-art models but not only the models proposed about 20 years ago.

**Questions:**

Please see the above limitations and address the questions. In addition to the limitations, another question is:

Based on Eqs. 9, 13, and 17, both the upper bound of the posterior violation probability and the allocable risk at specific time step are fixed for the average risk allocation setting, as the upper bound of the posterior violation probability is calculated based on the calibration data D_{cal}^{2}. This violates the real-world situation where the real-world data and calibration data follow different distribution. However, experiment result in Table 1 shows that this average risk allocation method still achieves 89.1% collision avoidance rate. Are there any data distribution shifts in the testing data? If not, what’s the performance of the proposed method under distribution shift?

---

> ### Author Response · Authors · 2024-11-21
> **Rebuttal by Authors – Part I**
>
> We thank the reviewer for the constructive comments! We have revised the manuscript to address the concerns.
>
> >**Weakness 1:** The prediction regions are expected to cover the ground truth trajectory and narrow. Otherwise, overlarge prediction regions would also achieve a high collision avoidance rate. The authors should provide more details about how narrow the prediction regions are.
>
> Thank you very much for your valuable comment. The proposed method actually provides theoretical guarantees for both the coverage and efficiency (narrowness) of the prediction region. On one hand, the coverage guarantee of the prediction region is provided in Lemma 3.1 of the original manuscript. On the other hand, Corollary 4.1 restricts the upper bound of the expectation of posterior probability. This ensures that the prediction region obtained by the proposed method is at least narrower than the one obtained through the state-of-the-art sequential method [1], and its efficiency can be continuously improved during trajectory optimization. In fact, the ability to achieve the narrowing of the prediction region while ensuring coverage is precisely the advantage and contribution of the proposed method.
>
> Following the reviewer’s comments, we provide more details about how narrow the prediction regions are through experiments. Table 8 (Appendix F) presents the prediction region radius for different time $t$ and $\tau$ ($C_{\tau|t}$) with different methods. Since iterative risk allocation affects the prediction region radius, we only compared $C_{\tau|t}$ obtained through E2E-CP with ARA and Sequential CP to make a fair comparison. At the initial state ($t=0$), no realized state is available for calculating posterior probabilities. As a result, $C_{\tau|0}$ for all $\tau$ obtained by Sequential CP and E2E-CP with ARA is essentially identical. As the system operates, an increasing number of realized states $x_{t}^{*}$ are available for the calculation of posterior probabilities, enabling E2E-CP to yield a relatively narrower prediction region, i.e. a smaller $C_{\tau|t}$. Moreover, when $t>10$, based on sufficient posterior probabilities, the prediction region radius obtained by E2E-CP with ARA is reduced by more than $50$% compared to that of Sequential CP. The reviewer can find a detailed discussion on the narrowness of the prediction region in Appendix F and lines 500 to 512 of the revised manuscript.
>
> [1] *Lars Lindemann, Matthew Cleaveland, Gihyun Shim, and George J Pappas. Safe planning in dynamic environments using conformal prediction. IEEE Robotics and Automation Letters. 2023.*

---

> ### Author Response · Authors · 2024-11-21
> **Rebuttal by Authors – Part II**
>
> >**Weakness 2:** The authors should demonstrate the model performance by implementing more state-of-the-art models but not only the models proposed about 20 years ago.
>
> Thank you very much for your valuable comment. First, we would like to clarify that the primary contribution of this paper is to propose the E2E-CP - a novel end-to-end Conformal Prediction (CP) method for Trajectory Optimization (TO), rather than a novel system model. Therefore, in the experiments of the original manuscript, we compared our method with the state-of-the-art CP-based TO method (2023) [1] to demonstrate the performance of our method.
>
> Additionally, the two robotic models employed in our experiments of the original manuscript—the kinematic vehicle model and the quadrotor model—are classical robotic models and serve as mainstream benchmarks for evaluating the performance of algorithms in trajectory optimization, trajectory planning, and motion control [1][2][3]. Many systems in daily life (car-like wheeled mobile agents, unicycles, etc) can be modeled as the kinematic vehicle model which is used in our paper.
>
> Finally, we fully agree with the reviewer that testing our algorithm on additional models would provide a more comprehensive evaluation of its performance. Following the reviewer’s valuable suggestion, we implemented our algorithm on the dynamic bicycle model (2021) [2], and the experimental results and details are provided in Appendix C.3 of the revised manuscript. The experimental results show that, under this model, the proposed method achieves at least a $40.6$% reduction in average cost compared to the state-of-the-art method, further demonstrating the superiority and generalizability of the proposed method. The reviewer can find the detailed results (Table 5) and the analysis of the newly added experiment in Appendix C.3 of the revised manuscript.
>
> [1] *Lars Lindemann, Matthew Cleaveland, Gihyun Shim, and George J Pappas. Safe planning in dynamic environments using conformal prediction. IEEE Robotics and Automation Letters. 2023.*
>
> [2] *Astghik Hakobyan and Insoon Yang. Wasserstein distributionally robust motion control for collision avoidance using conditional value-at-risk. IEEE Transactions on Robotics. 2021.*
>
> [3] *Bolun Dai, Rooholla Khorrambakht, Prashanth Krishnamurthy, Vinicius Goncalves, Anthony Tzes, and Farshad Khorrami. Safe navigation and obstacle avoidance using differentiable optimization based control barrier functions. IEEE Robotics and Automation Letters. 2023.*

---

> ### Author Response · Authors · 2024-11-21
> **Rebuttal by Authors – Part III**
>
> >**Question:** Based on Eqs. 9, 13, and 17, both the upper bound of the posterior violation probability and the allocable risk at specific time step are fixed for the average risk allocation setting, as the upper bound of the posterior violation probability is calculated based on the calibration data D_{cal}^{2}. This violates the real-world situation where the real-world data and calibration data follow different distribution. However, experiment result in Table 1 shows that this average risk allocation method still achieves 89.1% collision avoidance rate. Are there any data distribution shifts in the testing data? If not, what’s the performance of the proposed method under distribution shift?
>
> Thank you very much for your valuable comment. **In the experiments of the original manuscript, there are no distribution shifts in the test data**. We will first demonstrate that the assumption of no distribution shift is not restrictive and holds in many cases. Furthermore, we experimentally demonstrate that the proposed method exhibits a certain degree of robustness to moderate distribution shifts and can maintain safety and high performance in realistic scenarios.
>
> In Section 3.1, the assumption of no distribution shift between calibration trajectories and test trajectories is derived from Assumption 3.1 (lines 136-137) and Assumption 3.2 (lines 138-139). We will demonstrate that these two assumptions are not restrictive in practice as follows:
>
> 1. Assumption 3.1 posits that the system does not influence the real joint obstacle trajectory, which holds approximately in many robotic applications, e.g., autonomous vehicles behave in ways that result in socially acceptable trajectories that do not change the behavior of pedestrians [1].
> 2. Assumption 3.2 assumes the availability of the training and calibration datasets, which is also not restrictive in practice, as extensive data can be sourced from advanced high-fidelity simulators or robotic applications like autonomous vehicles, where datasets are increasingly accessible.
>
> Although many scenarios approximately satisfy our assumption of the same distribution, we acknowledge that the system state $x$ may change $\mathcal{D}$ during test time, e.g., when a robot is too close to a pedestrian. However, we demonstrate through analysis and experiments that the proposed method exhibits a certain degree of robustness to moderate distribution shifts and can maintain safety and high performance in realistic scenarios (beyond the same distribution).
>
> **The performance of the proposed method under distribution shift:**
>
> We have added experiments on the performance of our method under distribution shifts in Appendix D of the revised manuscript. Specifically, we generate test trajectories with varying degrees of distribution shifts by applying zero-mean Gaussian noise with varying covariances to the accelerations of the obstacles. The experimental results indicate that under moderate distribution shifts, the proposed method can still satisfy the total risk tolerance constraint while maintaining an average cost reduction of at least $57.1$%. The reviewer can find the detailed results (Table 6) and the analysis of the experiment in Appendix D of the revised manuscript.
>
> We would like to further clarify that when the distribution shifts become excessively large, the proposed method inevitably exceeds the total risk tolerance. However, reference [2] explicitly checked through experiments that interactions among agents in multi-agent systems do not introduce large distribution shifts. Furthermore, we directly design simulation scenarios where the obstacles’ avoidance behavior for the system (beyond Assumption 3.1) is incorporated. The experimental results indicate that the reduction in the collision avoidance rate due to the dependence between the system and obstacles is negligible. The reviewer can find the detailed results (Table 7) and the analysis of the experiment in Appendix E of the revised manuscript.
>
> In summary, the robustness of our method to distribution shifts is sufficient to handle realistic scenarios. Additionally, in Appendix G of the revised manuscript, we discuss several potential ways to address the distribution shift in full generality or to provide theoretical guarantees.
>
> [1] *Lars Lindemann, Matthew Cleaveland, Gihyun Shim, and George J Pappas. Safe planning in dynamic environments using conformal prediction. IEEE Robotics and Automation Letters. 2023.*
>
> [2] *Kegan J Strawn, Nora Ayanian, and Lars Lindemann. Conformal predictive safety filter for rl controllers in dynamic environments. IEEE Robotics and Automation Letters. 2023.*

---

> ### Author Response · Authors · 2024-12-01
> **Looking forward to your feedback. Many thanks!**
>
> Dear Reviewer xXLt,
>
> Again, thank you so much for your previous valuable comments. We would like to follow up on our rebuttal to ensure that all your concerns have been adequately addressed. If there are any further questions, we will be very glad to address them. Your feedback is extremely valuable in helping us improve our manuscript, and we eagerly await your reply.
>
> Thank you very much for your time and consideration.
>
> Best regards,
>
> The Authors

---

> > ### Comment · Reviewer_xXLt · 2024-12-03
> > **Reply to the Rebuttal**
> >
> > Thank you for the detailed response and the additional experimental results. We appreciate the time and effort. After careful consideration, I will maintain my original score.

---

> > > ### Author Response · Authors · 2024-12-03
> > > **Thank you so much for your reply**
> > >
> > > We sincerely thank you for the kind reply and providing valuable review comments which have helped us to improve the quality of this manuscript.

---

### Author Response · Authors · 2024-11-29
**General Response**

Dear Reviewers and ACs,

We sincerely thank all the reviewers and ACs for your time and efforts on providing valuable review comments and questions, which have helped us to improve the quality of this manuscript. If you have any additional questions or require further clarification, please feel free to let us know. Your insights are highly appreciated.

In our paper, we propose, for the first time in the literature E2E-CP, an end-to-end framework based on Conformal Prediction (CP) for shrinking-horizon Trajectory Optimization (TO) in uncertain environments. The proposed E2E-CP enjoys both performance and validity. Specifically, E2E-CP leverages CP to construct the prediction regions of obstacle positions and adjusts these regions online using the feedback information embedded in decisions. Additionally, such end-to-end adjustments in E2E-CP do not compromise the theoretical coverage guarantees of prediction regions. In the original manuscript, E2E-CP was validated on the kinematic vehicle model and the quadrotor model, demonstrating a reduction in average cost of over 78.5% and 58.0%, respectively, compared to the state-of-the-art method.

Most reviewers are concerned about the performance of our method across a wider range of models and under different scenarios, such as distribution shift and interdependence. So we conduct additional experiments and present the results in the appendix of the revised manuscript. The newly added content in the revised manuscript is highlighted in blue.

In the additional experiments, we implement our method on the dynamic bicycle model (2021) [1] in Appendix C.3, investigate the performance of our method under distribution shift in Appendix D, and examine the performance of our method under the interdependence between the system and obstacles in Appendix E. We provide a brief summary and discussion of the additional experimental results as follows. For detailed results and discussions, reviewers are kindly referred to the appendix of the revised manuscript.

In the experiments on the dynamic vehicle model [1], E2E-CP achieves at least a 40.6% reduction in average cost compared to the state-of-the-art method, further demonstrating the superiority and generalizability of E2E-CP.

In the experiments on distribution shifts, E2E-CP can still satisfy the total risk tolerance constraint while maintaining an average cost reduction of at least 57.1% under moderate distribution shifts. The experimental results demonstrate that E2E-CP exhibits a certain degree of robustness to moderate distribution shifts.

In the experiments on interdependent scenarios, the reduced collision avoidance rate caused by the interaction between the system and obstacles is negligible and remains well within the total risk tolerance. Moreover, the dependence between the system and obstacles has almost no impact on the average cost empirically. In summary, the interactions among agents in multi-agent systems do not significantly affect our algorithm and its performance.

Moreover, we provide more details about how narrow the prediction regions are in Appendix F, and discuss the limitations of our method in the main text.

We hope these new results and discussions can address the reviewers’ concerns.

[1] Astghik Hakobyan and Insoon Yang. Wasserstein distributionally robust motion control for collision avoidance using conditional value-at-risk. IEEE Transactions on Robotics. 2021.

Best regards,

The Authors

---

### Meta-Review · Area_Chair_xsEW · 2024-12-23

**Metareview:**

The paper proposes a method for online trajectory optimization under uncertainty by applying conformal prediction in a chance-constrained optimization framework. The experiments are basic and are a point of criticism by the reviewers. While two reviewers recommend acceptance and two reviewers think that the paper is marginally below acceptance, I am recommending rejection because the experiments are weak and the baselines are fairly old.

**Additional Comments On Reviewer Discussion:**

The authors provided detailed rebuttals - however the two most critical reviewers still decided to maintain their rating, while the most positive reviewer did not make an effort to argue or push for the paper.

---

### Decision · Program_Chairs · 2025-01-22

Reject